# Linking plasmid-based beta-lactamases to their bacterial hosts using single-cell fusion PCR

**Peter J Diebold[1], Felicia N New[1], Michael Hovan[2], Michael J Satlin[3], Ilana L Brito[1]***

[1]Meinig School of Biomedical Engineering, Cornell University, Ithaca, United States; [2]Robert Wood Johnson Medical School, New Brunswick, United States; [3]Weill Cornell Medicine, Cornell University, New York, United States

**Abstract** The horizontal transfer of plasmid-encoded genes allows bacteria to adapt to constantly shifting environmental pressures, bestowing functional advantages to their bacterial hosts such as antibiotic resistance, metal resistance, virulence factors, and polysaccharide utilization. However, common molecular methods such as short- and long-read sequencing of microbiomes cannot associate extrachromosomal plasmids with the genome of the host bacterium. Alternative methods to link plasmids to host bacteria are either laborious, expensive, or prone to contamination. Here we present the One-step Isolation and Lysis PCR (OIL-PCR) method, which molecularly links plasmid-encoded genes with the bacterial 16S rRNA gene via fusion PCR performed within an emulsion. After validating this method, we apply it to identify the bacterial hosts of three clinically relevant beta-lactamases within the gut microbiomes of neutropenic patients, as they are particularly vulnerable multidrug-resistant infections. We successfully detect the known association of a multi-drug resistant plasmid with *Klebsiella pneumoniae*, as well as the novel associations of two low-abundance genera, *Romboutsia* and *Agathobacter*. Further investigation with OIL-PCR confirmed that our detection of *Romboutsia* is due to its physical association with *Klebsiella* as opposed to directly harboring the beta-lactamase genes. Here we put forth a robust, accessible, and high-throughput platform for sensitively surveying the bacterial hosts of mobile genes, as well as detecting physical bacterial associations such as those occurring within biofilms and complex microbial communities.

*For correspondence:
ibrito@cornell.edu

Competing interests: The authors declare that no competing interests exist.

## Introduction

The emergence of multidrug-resistant (MDR) pathogens is a grave public health threat that occurs when pathogenic bacteria acquire antibiotic-resistant genes (ARGs) through horizontal gene transfer (HGT) with bacteria in their proximal environment. The gut microbiome harbors a diverse repertoire of ARGs, and these genes have been proposed to serve as a reservoir for HGT with MDR pathogens (*Sommer et al., 2009*). ARGs are often carried on mobilizable plasmids that impose technical challenges to surveying the set of bacteria affiliated with these genes. Standard molecular tools such as PCR and next-generation sequencing often fail to associate mobile ARGs with their bacterial hosts because they cannot capture the cellular context of extrachromosomal genes. Novel untargeted sequencing methods, such as bacterial Hi-C (*Kent et al., 2020*) and methylation profiling (*Beaulaurier et al., 2018*), provide broad reconstruction of plasmid–host relationships in metagenomes, as a trade-off for sensitivity. Alternatively, single-cell whole-genome sequencing offers an ideal solution to this problem, but may be lower throughput, more expensive and require specialized equipment (*Xu et al., 2016*; *Lan et al., 2017*). Targeted methods, such as bacterial cell culture under antibiotic selection, require that the ARG is expressed, functional, and selective in all hosts.

Culturing, applied broadly to capture the full diversity of the gut microbiome, is complicated by the need for wide-ranging media and growth conditions (*Zou et al., 2019*; *Poyet et al., 2019*).

Several targeted methods using single-cell qPCR have been used to identify the hosts of specific genes; however, each uses specialized microfluidic devices, is limited in bacterial taxa they can capture, and most do not allow direct sequencing of the PCR products (*Ottesen et al., 2020*; *Zeng et al., 2010*; *Tadmor et al., 2011*). Alternatively, epicPCR (*Spencer et al., 2015*) uses fusion PCR and two emulsion steps to associate a taxonomic marker with a functional gene. Sequencing the fused PCR products provides accurate and sensitive associations between 16S sequence taxonomy and a given target gene. However, this method can be challenging to execute, difficult to scale up for multiple samples, and utilizes toxic and difficult-to-acquire reagents.

Here, we put forth One-step Isolation and Lysis PCR (OIL-PCR), a method that detects host–ARG associations from complex microbial communities through cellular emulsion and fusion PCR. Our streamlined method, based on the innovation of epicPCR, simplifies the procedure by combining the two emulsion steps of cell lysis and fusion PCR into a single emulsion PCR that can be performed in a 96-well format using robotic automation. Furthermore, OIL-PCR can be multiplexed to target at least three genes in the same reaction, uses non-toxic commercially available reagents, and can be performed without relying on microfluidics or specialized equipment. Validation experiments on three environmental bacterial communities reveal that OIL-PCR is highly accurate and specific. We demonstrate the utility of this approach in examining the bacterial hosts of three extended spectrum beta-lactamase genes in the gut of neutropenic patients.

## Results

### Development of the OIL-PCR method

OIL-PCR applies established fusion PCR methods to fuse any gene of interest to the 16S rRNA gene using three primers: two primers hybridize to the target gene and a universal 16S reverse primer hybridizes to the V4 region. Amplification of the target gene appends a universal 16S forward primer sequence to the end of the target amplicon via a tailed reverse primer. The target gene amplicon then acts as a primer for amplification and hybridizes to the 16S rRNA gene as a forward primer, producing a fused gene product containing both the target gene and the 16S V4 sequence (*Figure 1a*, *Figure 1—figure supplement 1*).

For fusion PCR to accurately link target genes with host marker genes, cells must be isolated to prevent the formation of non-specific fusion products. Oil emulsions and microwells have long been used to isolate eukaryotic cells; however, it is difficult to lyse bacteria in this format, especially gram-positive bacteria due to their thick cell walls. Existing single-cell isolation methods for bacteria either do not address this problem (*Zeng et al., 2010*; *Tadmor et al., 2011*), rely on specialized microfluidics (*Liu et al., 2018*), or use time-consuming methods to encapsulate bacteria within hydrogel beads before performing multi-step chemical and enzymatic lysis procedures (*Spencer et al., 2015*; *Tamminen and Virta, 2015*). To address this problem, OIL-PCR combines bacterial isolation, lysis, and fusion PCR into a single streamlined reaction.

We developed a protocol that allows for the incorporation of Ready-Lyse (RL) Lysozyme into the fusion PCR master mix. Whole bacterial cells are added directly to the master mix while on ice to inhibit lytic activity during sample preparation. Vigorous shaking of the mixture then encapsulates the individual cells in an emulsion. Warming the emulsion to 30℃ activates the enzyme, lysing the cells. Next, a standard PCR thermocycler carries out the fusion PCR in the single-cell emulsions. Fused PCR products are purified from the emulsion and amplified further with a nested primer to filter out off-target PCR products and add Illumina adapters. Lastly, custom indexing primers are used to index the fused products before Illumina sequencing. Our experiments confirmed the compatibility of the RL Lysozyme with the fusion PCR, but required the addition of bovine serum albumin (BSA), a globular protein known to reduce protein aggregation (*Finn et al., 2012*; *Figure 1—figure supplement 2a*). We found that RL retained full activity in the standard NEB Phusion HF buffer (*Figure 1—figure supplement 2b*).

Next, we optimized the fusion PCR master mix to maintain a stable emulsion and amplify efficiently in picoliter droplets. PCR emulsions were prepared with fluorinated oil as used in modern emulsion-based methods, such as Drop-Seq (*Macosko et al., 2015*) and digital qPCR (BioRad

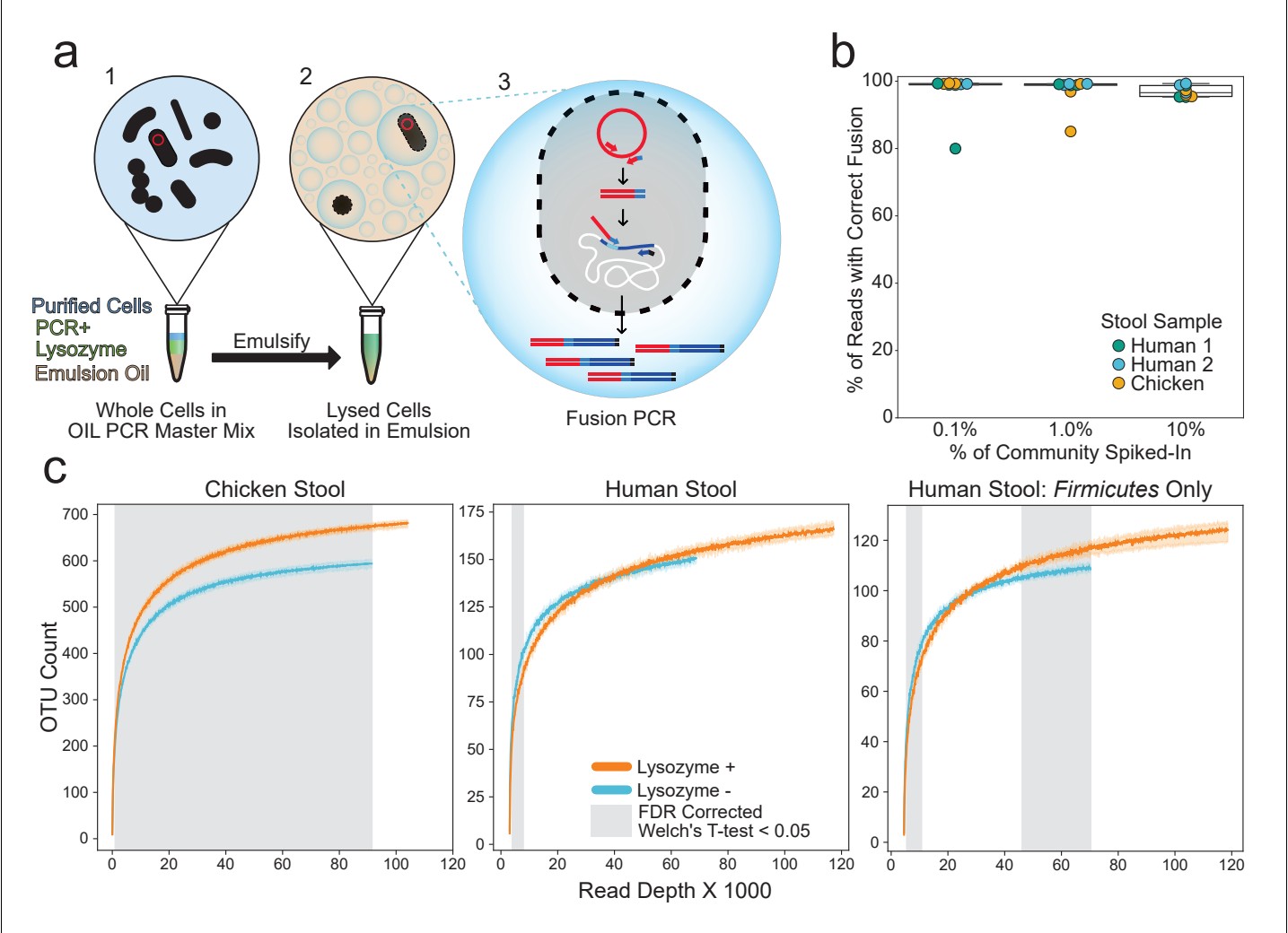

**Figure 1.** OIL-PCR can specifically link plasmid-encoded genes with their hosts. (a) Depiction of the OIL-PCR method. (1) Nycodenz-purified cells are mixed with PCR master mix, lysozyme, and emulsion oil and shaken to create an emulsion. (2) Cells are lysed within the emulsion. (3) Fusion PCR is performed in droplets containing cells harboring the targeted gene. Fused amplicons between the gene of interest and the 16S rRNA gene are the product. (b) A boxplot showing the percent of Illumina reads containing correct fusion products, namely the fusion of plasmid-borne *cmR* and the 16S rRNA gene of *E. coli* MG1655. OIL-PCR was performed on two individuals' and one chicken's gut microbiome sample in triplicate, spiked with varying concentrations of *E. coli*. (c) Rarefaction analysis of chicken (left) or human gut microbiome sample (middle) with (orange) and without (blue) lysozyme treatment. At right is the rarefaction analysis performed on Firmicutes only in the human stool sample. Grayed regions in the plot represent areas where the curves, each composed of four technical replicates, are significantly different (p<0.05) from one another, according to an FDR-corrected Welch's t-test.

The online version of this article includes the following figure supplement(s) for figure 1:

**Figure supplement 1.** Depiction of the fusion PCR.

**Figure supplement 2.** BSA and excess MgCl$_2$ improve the efficiency of OIL-PCR and Ready Lyse Lysozyme remains active in OIL-PCR master mix.

**Figure supplement 3.** Cell concentration of 400 cells/µl, DNase treatment, and multiplexing PCR result in accurate OIL-PCR results.

**Figure supplement 4.** Computational workflow.

**Figure supplement 5.** Lysozyme alone improved recovery of species.

**Figure supplement 6.** Lysozyme treatment improves OTU capture of most taxonomic groups in chicken and human stool.

**Figure supplement 7.** Combining replicates for increased depth improved recovery of species and reduced stochastic sampling bias.

ddPCR). We combined the fusion PCR master mix with bacterial cells and emulsion oil in either a 1.5 ml tube or a 0.5 ml deep-well plate before emulsifying the mixture using a tabletop bead homogenizer. Unlike microfluidic-enabled emulsions, our protocol leverages equipment commonly found in most molecular biology laboratories. We stabilized the emulsion with detergent-free buffers and

improved the efficiency of the PCR amplification within the emulsion by adding additional polymerase, BSA, dithiothreitol (DTT), and ammonium sulfate. We found that the addition of extra $MgCl_2$ mitigated the inhibitory effects of extremely high concentrations of cell debris within droplets after lysis (*Figure 1—figure supplement 2c*).

## OIL-PCR accurately associates plasmid genes with the host in a binary community

In any emulsion-based method, it is essential to optimize the concentration of input cells to prevent the encapsulation of two or more cells in the same droplet. When using a monodisperse emulsion such as those generated using microfluidics, the ideal concentration of input cells is chosen using a Poisson distribution (*Ottesen et al., 2020*; *Zeng et al., 2010*; *Tadmor et al., 2011*). However, these calculations are not reliable in the case of a polydisperse emulsion, as employed here to avoid the need for microfluidic devices. We therefore developed a probe-based TaqMan qPCR assay to experimentally verify the optimal concentration of input cells that prevented non-specific gene fusions (*Figure 1—figure supplement 3a*). OIL-PCR was performed on a binary community consisting of *E. coli* carrying the chloramphenicol resistance gene *cmR* on a plasmid and WT *V. cholerae*. The two strains were mixed 1:1, and we performed OIL-PCR with a fusion primer set specific to *cmR* and universally targeting the 16S rRNA gene (*Spencer et al., 2015*; *Supplementary file 2* and *3*). A gradient of cell input concentrations was used, and the final PCR products were recovered and purified. We then performed probe-based qPCR on the purified product using a nested primer for *cmR*, two blocking primers to inactivate any unfused amplicons, and two distinct fluorescent TaqMan probes (Thermo Fisher 4316034) to specifically target the V4 region of either *E. coli* or *V. cholerae* (*Supplementary file 2*). The fluorescent signal from each probe measured the relative ratio of specific to non-specific gene fusions present in the final amplicon pool (*Figure 1—figure supplement 3a*). When the input concentration of cells was at or lower than 400 cells/µl, or 40 k cells per reaction, non-specific gene fusion detection was reduced to undetectable levels (*Figure 1—figure supplement 3b*). As well as confirming that bacterial cells were isolated within the emulsion, we further confirmed that droplets did not coalesce by performing the TaqMan assay on OIL-PCR products from *E. coli* and *V. cholerae* cells combined after they were individually emulsified (*Figure 1—figure supplement 3b*). Our results confirmed that the emulsion is highly stable and coalescence was undetected.

## Application of OIL-PCR to environmental microbial communities allows robust and sensitive association of extrachromosomal elements with their host

Using OIL-PCR on environmental microbial communities requires clean bacterial cell preparations free of environmental contaminants, which may inhibit PCR. To address this concern, cells were purified using Nycodenz density gradient centrifugation (*Holmsgaard et al., 2011*; *Hevia et al., 2015*), a simple method that can isolate clean bacterial fractions with minimal handling time to reduce contamination. Additionally, concerned that cell-free DNA can stick to the membranes and cell walls of bacteria (*Vorkapic et al., 2016*), thus introducing noisy associations in the data, we treated cells with heat-liable double-strand-specific DNase (dsDNase). This enzyme only digests unprotected double stranded genomic DNA present in the samples without degrading single-strand primers. By controlling the enzyme concentration, temperature, and speed at which cells were processed, we were able to digest extra-cellular DNA without impacting PCR efficiency of cellular contents. Using our Taqman assay, we demonstrated that including dsDNase treatment has the potential to increase the total cell input per reaction tenfold (*Figure 1—figure supplement 3c*).

To test the accuracy of our method on environmental samples, we spiked *Escherichia coli* MG1655 (*Blattner et al., 1997*) containing plasmid pBAD33 (*Guzman et al., 1995*) harboring the *cmR* gene into two human and one chicken stool samples that lacked the gene according to PCR screening. We performed OIL-PCR in triplicate with primers targeting the *cmR* gene and sequenced using MiSeq 2x250 reads. Paired-end reads were merged and quality filtered before splitting them at the fusion primer junction. The target portion of each read was confirmed to match the *cmR* gene and taxonomy was assigned to the 16S portion of each read (*Figure 1—figure supplement 4*). Our results show that when *E. coli* was incorporated at 0.1%, or about 20 cells total, 97.8% of the reads

(or 99.2%, excluding a single outlier) demonstrated the correct association when the test strain of *E. coli* was incorporated at 0.1%, or about 20 cells total (*Figure 1b*), highlighting the sensitivity of OIL-PCR to detect the associations of genes in low-abundant species across different sample types. The accuracy of OIL-PCR decreases slightly when the targeted sequence increases to 10% of the community composition, although associations were still 97% correct on average.

## Lysozyme improves capture of difficult-to-lyse gram- positive bacteria

To achieve our goal of robust lysis and amplification to screen all bacteria within a complex community, we measured the effect lysozyme had on bacterial detection. We performed standard 16S sequencing on human and chicken stool communities using OIL-PCR, testing three variables: the effect of lysozyme, dsDNase, and heat inactivation of dsDNase on total bacterial recovery (*Figure 1—figure supplement 5*). All eight combinations of the three variables were tested in duplicate for two stool samples using robotic automation. For our analysis, we chose to focus on the total number of operational taxonomic units (OTUs) captured in our data rather than relative abundance metrics, as this better reflects our goal of detecting species, rather than recapitulating the starting community structure.

First, we assayed how each of the three variables (RL, dsDNase, and heat inactivation) affected OTU recovery. Based on rarefaction curves, we found dsDNase and heat inactivation had no significant effect on OTU recovery in human and chicken stool, while RL lysozyme significantly increases OTU recovery in chicken stool based on Welch's t-test with Benjamini–Hochberg FDR correction (*Figure 1c*, *Figure 1—figure supplement 5*). RL was the only variable that significantly changed OTU recovery, and therefore, it was the only variable considered for further analysis.

Next, we looked to see which taxonomic groups were being enriched or depleted with the addition of lysozyme. Technical replicate OTU tables were combined for analysis to allow for deeper sampling depth. Rarefaction curves were generated for both lysozyme treatments at each taxonomic level containing 10 or more OTUs from Phylum to Genus. Results show that no taxonomic group was significantly depleted in either human or chicken stool samples (*Figure 1—figure supplement 6*). Chicken rarefaction curves trended higher with lysozyme for every taxonomic group tested, with significant improvements for the phyla *Firmicutes*, *Bacteroidetes*, and *Cyanobacteria* (*Figure 1—figure supplement 6*). Overall, 14 taxonomic groups were significantly enriched in chicken stool, mostly from *Firmicutes* and *Bacteroidetes*.

The effect of lysozyme on human stool was not as pronounced as for chicken stool, but it did significantly enrich for the *Firmicutes* phylum as well as the *Lachnospiraceae* family. The only group that trended worse with lysozyme was the *Bacteroidetes* (p-val 0.45), with the family *Bacteroidaceae* accounting for most of the effect. Interestingly, the closely related and biologically important family *Prevotellaceae* was enriched with a p-val of 0.07 (*Figure 1—figure supplement 6b*). While we cannot fully explain why lysozyme generally improves capture of *Bacteroidetes* in chicken but not human stool, the overall benefit of lysozyme is apparent, especially for capturing the breadth of diversity within the *Firmicutes* phylum. We noticed that the total number of OTUs recovered from OIL-PCR was significantly lower than 16S sequencing of the input community at the same sampling depth (*Figure 1—figure supplement 7*). We hypothesized the reason for this reduction in OTUs was due to subsampling bias introduced through low cell input and variable amplification efficiency in OIL-PCR. To test our hypothesis, we combined OTU tables from two, four, and eight technical replicates and found a consistent up-shift for each rarefaction curve as we combined more tables. This up-shift was not observed when combining the input Nycodenz sequencing, indicating that the reduced OTU counts were due in part to subsampling bias and not an inherent failure to capture bacterial taxa (*Figure 1—figure supplement 7*). We therefore recommend OIL-PCR to be performed in replicates to increase the total number of cells being sampled.

## Increased throughput through automation and multiplexing

To further improve the efficiency and throughput of OIL-PCR, we sought to transition the method from 1.5 ml centrifuge tubes to a 96-well plate format using the Eppendorf epMotion liquid handling robot. The liquid handling robot can perform certain parts of the PCR preparation as well as DNA recovery and purification. The automated workflow allowed us to process up to 48 samples simultaneously with fewer manual steps overall.

We next tested whether OIL-PCR could simultaneously target multiple genes though multiplexing. We repeated the previously described TaqMan assay using a strain of *V. cholerae* containing the ampicillin resistance gene *ampR* and *E. coli* with *cmR*, both on a plasmid (*Figure 1—figure supplement 3d*). Our results demonstrate that OIL-PCR can be multiplexed while still accurately maintaining the correct associations of target genes with their host bacteria.

## Bacterial hosts are identified for several clinically important β-lactamase genes

We analyzed metagenomic sequencing of stool samples that were collected from a cohort of patients who were neutropenic because of chemotherapy administered for a hematopoietic cell transplant. Two patients, B335 and B314, were chosen for OIL-PCR based on the presence of three class-A beta-lactamase genes, $bla_{TEM}$, $bla_{SHV}$, and $bla_{CTX-M}$ in the metagenomes (*Kent et al., 2020*). We tested a three-sample time course from patient B335: before antibiotic treatment, after 4 days of trimethoprim-sulfamethoxazole and 1 day of levofloxacin, and lastly after an additional 2 days of levofloxacin (*Figure 2a*). Patient B335 carried all three genes across three time points with $bla_{TEM}$ and $bla_{CTX-M}$ on a metagenomic scaffold which blasted to an 80 kb *Klebsiella* plasmid and $bla_{SHV}$ on a contig that blasted to *K. pneumoniae* genome (*Figure 2b*). Previously published Hi-C sequencing of the stool samples identified an association between *K. pneumoniae* and the 80 Kb plasmid, as well as transfer to *Citrobacter brakii* between time points 1 and 2 (*Kent et al., 2020*). We tested one sample from patient B314 from before antibiotic treatment which carried multiple $bla_{SHV}$ genes. We hypothesized that OIL-PCR could be used to sensitively and accurately detect additional hosts of these genes.

We designed three degenerate fusion primer sets to broadly target most variants of $bla_{TEM}$, $bla_{SHV}$, and $bla_{CTX-M}$ (*Supplementary file 2* and *3*), and performed multiplexed OIL-PCR with robotic automation. Samples were processed in quadruplicates. We set a threshold for defining positive gene–taxa associations, as having 0.5% of total reads across the four technical replicates.

Our OIL-PCR results largely confirm findings in the metagenomic assemblies from *Kent et al., 2020*. In B314, we found $bla_{SHV}$ associated with *Klebsiella* as suggested by metagenomic assemblies. However, we also detected two other class-A beta-lactamase genes, $bla_{LEN}$ and $bla_{OXY}$, which were present in the metagenomes, but we did not expect to amplify with our primers. $bla_{LEN}$ amplified with the primers designed for $bla_{SHV}$ and $bla_{OXY}$ amplified with primers for $bla_{CTX-M}$. Curiously, $bla_{OXY}$ is an exceptionally poor match for our $bla_{CTX-M}$ primers, having a mismatch one base away from the 5' end of the fusion primer. We hypothesize that the low annealing temperature and modified buffer used in the emulsion PCR is highly permissive to priming mismatches. We see permissive annealing as an advantage for the method because it allows for amplification of unknown variants of target genes while amplification due to off-target priming is filtered out during the nested PCR step (*Figure 1—figure supplement 1*), leaving only the true amplicons in the final sequencing. This permissive annealing behavior of OIL-PCR can be leveraged in the future to design broad-range primers for diverse gene groups such as metallo-beta-lactamases (*Somboro et al., 2018*).

Results from patient B335's time course also matched the metagenomic and Hi-C sequencing from Kent et al., associating $bla_{TEM}$, $bla_{SHV}$, and $bla_{CTX-M}$ with *Klebsiella* in all three time points (*Figure 2c,d*). We also found that all three genes strongly associated with the commensal genus *Romboutsia* in time points T2 and T3 and to a lesser extent with *Agathobacter* in time point T1 (*Figure 2c,d*). *Citrobacter brakii*, which was detected as a recipient of the *Klebsiella* plasmid in the Hi-C sequencing, did not initially show up in our analysis as it was clustered with *Klebsiella*, as its 16S sequence differs by only a single base pair. However, upon closer inspection, the *C. brakii* strain does appear to be associated with the three genes in time points 2 and 3 only. These results indicate that, using manual inspection or by modifying our computational pipeline, higher resolution associations can be obtained by OIL-PCR. A strain of *Escherichia* with a distinct variant of $bla_{TEM}$ was detected at time point T2, but did not pass the detection threshold across all replicates in time point T3. We repeated OIL-PCR on all three samples from B335, this time in triplicate without multiplexing to further confirm these results. The singleplex experiment perfectly mirrored the multiplex results, excluding one replicate of T2/CTX-M which failed to sequence, indicating that these genes may be linked with organisms other than *Klebsiella*. As further confirmation of this result, we targeted two Tn3-like transposon genes situated in close proximity to $bla_{TEM}$ and $bla_{CTX-M}$ on the 80 kb *Klebsiella* plasmid. We hypothesized that these genes should also be associated with the same

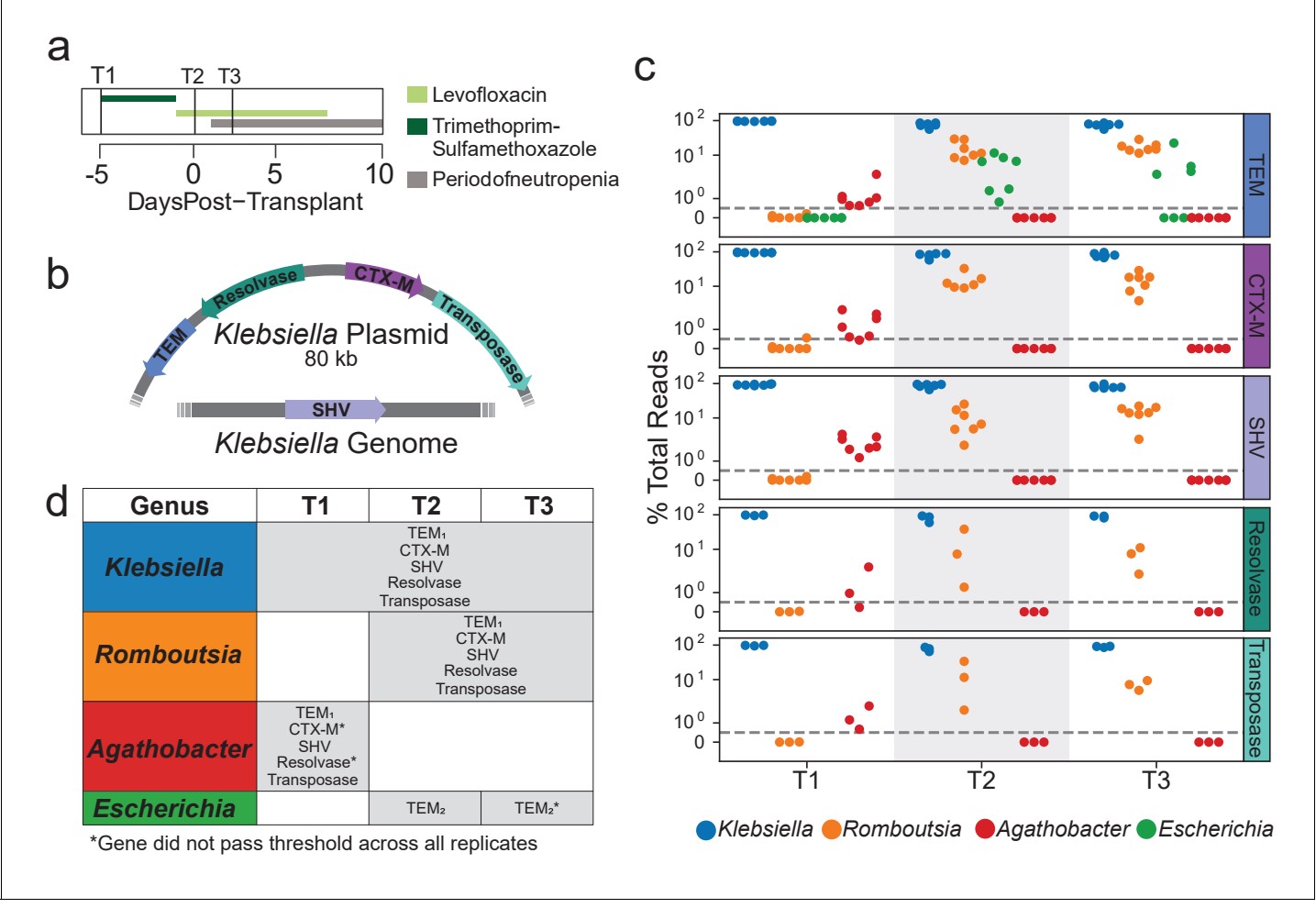

**Figure 2.** Extended spectrum beta-lactamase genes are associated with both pathogenic and commensal species. (**a**) Summary of treatment and sample time points for patient B335. (**b**) Depiction of an 80 kb plasmid carried by K. pneumoniae harboring the blaCTX-M, blaTEM, Tn3 transposase and resolvase genes. The blaSHV gene is presumed to be carried within the K. pneumoniae genome. Placement of these genes was inferred from metagenomic assemblies of patient B335's gut microbiome sample. (**c**) OIL-PCR results for each of the genes depicted in (**a**) patient B335 at three time points. For all gene-taxa associations, the percent of total OIL-PCR reads for that gene-time point is plotted. All species passing our detection threshold of 0.5% (dotted line) at any of the three time points is included in this plot. (**d**) A table summarizing the results in (**b**). All gene-taxa associations for each time point passing our detection thresholds are listed. Two SNP variants of TEM were detected and denoted with subscript numbering. Gene-taxa associations which did not consistently pass our detection threshold across all technical replicates are noted (*).
The online version of this article includes the following figure supplement(s) for figure 2:

**Figure supplement 1.** Species targeted fusion primers reveal the physical association of *Klebsiella* with *Romboutsia*.

genera as the ARGs. Remarkably, we observed the identical pattern with *Klebsiella, Romboutsia,* and *Agathobacter* as with the three beta-lactamases, but not *Escherichia*, which carried a distinct variant of bla$_{TEM}$ (*Figure 2c,d*).

## OIL-PCR provides further evidence of the association of beta-lactamases with the commensal *Romboutsia*

We next investigated whether OIL-PCR could be used to further confirm the association between *Romboutsia* and the three beta-lactamases. We focused specifically on *Romboutsia* because of the strong signal in the OIL-PCR results compared to *Agathobacter*. For this experiment, instead of fusing the ARG sequence to the 16S rRNA gene using universal primers, we used primers designed to specifically detect the *Romboutsia* 16S rRNA (*Gerritsen et al., 2014*) and fused the 16S gene specifically to bla$_{TEM}$ (*Figure 3a, Supplementary file 2* and *3*). In this instance, no amplification is possible

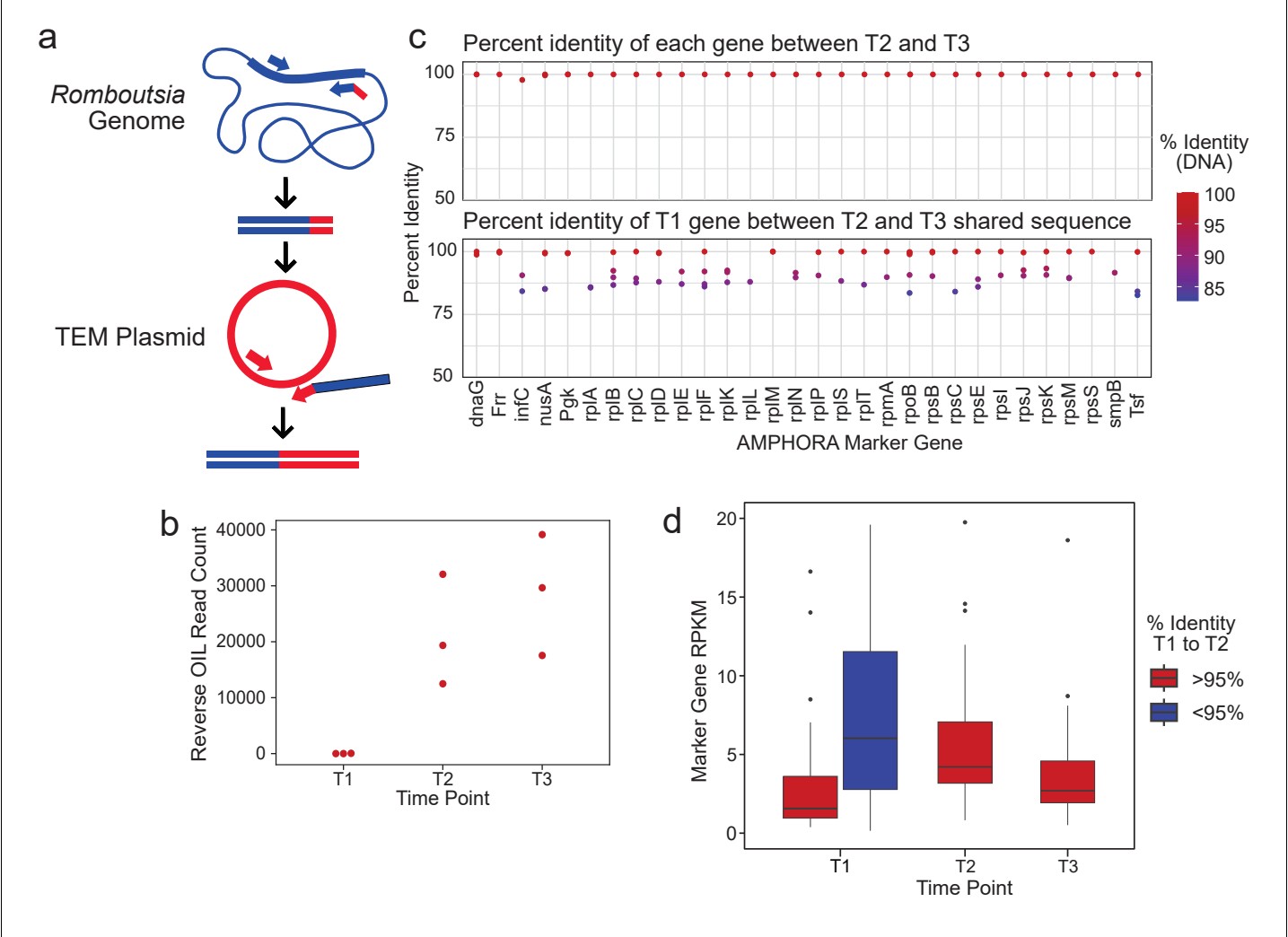

**Figure 3.** R.timonensis strains associated with the three beta-lactamase genes appear over the patient's time course. (a) Depiction of the reverse OIL-PCR in which *Romboutsia*-specific 16S rRNA sequences (blue) are fused with the *bla*$_{TEM}$ sequences (red). (b) OIL-PCR read counts of the reaction shown in (a) are plotted. (c) The percent sequence identity of assembled *R. timonensis* marker genes between genes identified in timepoints 2 and 3 (top) and between timepoint 1 and between sequences shared at timepoints 2 and 3 (bottom). (d) RPKM-normalized abundance-values for the assembled marker genes for each strain assembled in time point one and the major strain present in timepoints 2 and 3.

unless *Romboutsia* is encased in the same droplet with the *bla*$_{TEM}$ gene and would negate the possibility of false-positive associations due to chimera formation. Results show amplification and sequencing was only produced from time points T2 and T3 with no signal detected at time T1, confirming the presence of *bla*$_{TEM}$ within *Romboutsia* at time points T2 and T3, but not T1 (*Figure 3b*).

We next explored the metagenomic data for clues as to whether the *Romboutsia* strain was present at timepoint T1, but below the detection threshold, or whether the strain linked with the genes was acquired sometime between time T1 and T2. Based on the 16S data from OIL-PCR and metagenomic sequencing, we identified the *Romboutsia* species as *R. timonensis*. Genus-level abundance data showed *R. timonensis* to be present in all three timepoints in patient B335. Due to the overall low abundance of this organism, we were unable to assemble a *Romboutsia* genome from these samples. Instead, we aligned patient B335's three samples to the *R. timonensis* (PRJEB14233) genome from NCBI, assembled the aligned reads, and examined similarities between the *R. timonensis* taxonomic markers over the three timepoints (*Figure 3c*). We found that B335 was colonized by at least two independent strains of *R. timonensis* during the first timepoint, but that only one *R. timonensis* strain persisted during timepoint T2 and T3. One of the *R. timonensis* strains from T1 was

identical to the strain from T2 and T3 across 15/30 AMPHORA marker genes, and >99% identical in 24/30 genes (*Figure 3c*), suggesting that the strain of *R. timonensis* from time T2 and T3 was also present at time point T1. We found no significant difference in the normalized abundance of *Romboutsia* between time point T1 and time points T2 and T3 (*Figure 3d*), albeit our data suggests that the persistent strain is the minor variant at time point 1. Despite the sensitivity of OIL-PCR, which can detect cells at least 0.1% abundant (*Figure 1b*), we cannot rule out the possibility that the stochasticity of sampling in OIL-PCR and the low abundance of this particular strain of *R. timonensis* precluded our ability to observe this association at the beginning of the time course.

## OIL-PCR confirms the physical association between *Romboutsia timonensis* and *Klebsiella pneumoniae*

Our analysis clearly shows an association between *Romboutsia* and the three beta-lactamase genes; however, there are two plausible explanations for these results. Either *Romboutsia* acquired all three genes from *Klebsiella* through HGT, or *Romboutsia* and *Klebsiella* are physically linked together, causing them to consistently emulsify within the same droplet, thus allowing the *Romboutsia* 16S gene to fuse with the three ARGs. To distinguish between these two possibilities, we designed OIL-PCR primers targeting two *Klebsiella pneumoniae* housekeeping genes *rpoB* and *glmS*, and two *Romboutsia timonensis* genes *rpoB* and *nusA*. Using these primer sets, along with $bla_{CTX-M}$ primers as a control, we ran OIL-PCR to see if we found *Klebsiella* marker gene sequences fused to *Romboutsia* 16S or vice versa, suggesting capture of the two species within the same droplet. *Klebsiella* primers were multiplexed with $bla_{CTX-M}$ and the *Romboutsia* primers were assayed in separate reactions to rule out the possibility of PCR chimeras during library preparation.

The results show that both *Klebsiella* and *Romboutsia* 16S sequences were fused to both of the *Klebsiella* marker genes, mirroring the same pattern as was seen for the ARGs and transposase genes assayed. The $bla_{CTX-M}$ control also presented the same pattern as previously demonstrated (*Figure 2—figure supplement 1*). This result indicates that *Klebsiella* and *Romboutsia* are being emulsified together, suggesting a physical association and not gene transfer. Based on these results, we would also expect to also find the *Klebsiella* 16S sequence fused to the *Romboutsia* marker genes. Primers targeting *nusA* failed to amplify; however, the *Romboutsia*-specific *rpoB* primers did fuse to *Klebsiella* 16S sequences in two of the nine total replicates across three time points. These results, taken with our previous OIL-PCR experiments, present compelling evidence that the observations can be explained as a novel physical association between *Klebsiella pneumoniae* and *Romboutsia timonensis* that developed between strains present after time point 2 in patient B335.

## Discussion

Here we show the ease with which OIL-PCR can identify carriers of known resistance markers on extrachromosomal elements within complex bacterial communities. We applied it to a neutropenic patient's gut microbiome and showed the correct association of three beta-lactamases with *K. pneumoniae*, and also discovered novel associations between these beta-lactamases and two gut commensals, *R. timonensis* and *Agathobacter* spp. Two of the genes, $bla_{CTX-M}$ and $bla_{TEM}$, were both found on a large *Klebsiella* plasmid within the metagenome, suggesting the possible transfer of these genes to *R. timonensis* during the time course. However, analysis of the plasmid sequence showed that while it does contains an origin of transfer, it does not have the genes necessary to transfer itself, meaning it would require a second 'helper plasmid' to mobilize. Additionally, $bla_{SHV}$ was only found on a contig belonging to the *Klebsiella* genome without any known mobilizable transposons or integrative conjugative elements nearby, severely limiting its transfer potential. An alternative explanation for our results is that *Romboutsia* and *Klebsiella* became physically associated within the gut, and thus consistently emulsified together. Using OIL-PCR targeting species-specific marker genes, we showed that our results were indeed due to a novel physical interaction between *K. pneumoniae* and *R. timonensis*.

Whether mobilization of ARG-containing plasmids, or novel physical associates within the gut, our results highlight the strength of OIL-PCR for unraveling the intricate dynamics of the gut microbiome. The ability of OIL-PCR to detect two kinds of ecologically and clinically important interactions, as well as distinguish between them, is a major strength of the method. Additionally, both of these interaction types are deeply entwined, with close physical association being a known activator

for conjugal transfer of genes (*Clark et al., 2018*), as well as a mechanism for resistance in multispecies biofilms (*Burmølle et al., 2014*) OIL-PCR is a practical and transportable protocol with no requirements for specialized equipment nor specialized expertise. We identify improvements in performing single-cell analysis on stool, namely the use of a Nycodenz purification step and the incorporation of lysozyme plus heat-induced lysis. Additionally, we increased throughput at least threefold through primer multiplexing and developed an automated protocol to process at least 48 samples concurrently, allowing a total of 144 gene-sample association tests per batch.

Additional improvements to OIL-PCR could be explored to further increase throughput and sensitivity. Although we tested multiplexing three genes per reaction, this number could likely be increased as we have found no sign of false positives due to multiplexing as demonstrated by associating a novel $bla_{TEM}$ variant with only *Escherichia* in time point T2 of patient B335 (*Figure 2b,c*). Furthermore, we show that the OIL-PCR master mix facilitates permissive annealing of primers, allowing a mismatch one base from the primer's 3' end as demonstrated when $bla_{OXY}$ was detected in sample B324-2 with $bla_{CTX-M}$ primers. These results could allow for the development of highly degenerate primers to target a broad range of gene variants. Non-specific priming during OIL-PCR is not of concern because the nested PCR specifically filters out undesired fusion products. Because OIL-PCR uses three primers for each target gene, primers designed for OIL can easily be adapted for probe-based qPCR pre-screening of samples instead of using metagenomic sequencing as was done in this study.

While the startup cost of using OIL-PCR is low compared to other methods, currently it uses large amounts of Phusion polymerase and magnetic beads for DNA purification which inflates the cost (~$15/100 μl reaction). The amount of Phusion needed could be reduced with further optimization of the PCR master mix, and the number of purification steps can be cut by using enzymatic exonuclease I treatment of PCR instead of purification. Lastly, the method described currently allows 40,000 cells total per reaction; however, our probe-based qPCR assays suggest that the input concentration could be increased 10-fold by pretreating cells with dsDNase (*Figure 1—figure supplement 3c*). Combined with our result showing that OIL-PCR is more accurate when detecting low-abundant taxa (*Figure 1b*), we feel confident that cell input can be increased to improve sensitivity without sacrificing accuracy.

OIL-PCR is a highly versatile platform that could be applied across fields to address a multitude of questions. While we were interested in plasmid-born ARGs in the gut, the method could be used to target any gene of interest that is difficult to associate with a host using metagenomics. As mobile genetic elements are notoriously difficult to assemble due to their promiscuity which complicates de Bruijn graph assembly (*Antipov et al., 2019*), this method could be applied to find the hosts of integrated and non-integrated mobile elements. Similarly, as metavirome sequencing has revealed a massive number of viral genomes with unknown hosts (*Shkoporov and Hill, 2019*), OIL-PCR may be particularly useful in addressing this gap in understanding. Additionally, viral and plasmid host-range is an important determinant for understanding and modeling bacterial ecology of predation and HGT (*Flores et al., 2011*). As we have shown here, OIL-PCR can detect direct physical associations of bacteria. Such interactions are important for understanding biofilm composition (*Shi et al., 2020*), identifying endosymbionts, or detecting cross-feeding bacteria which require the direct exchange of nutrients to grow (*D'Souza et al., 2018*; *Goyal et al., 2021*), information which could allow for culturing of these often unculturable species. In cases when physical associations are not of interest, samples may be filtered to remove clumped bacteria. Furthermore, targeting functional metabolic genes detected in metagenomes, but present at low abundance in bacterial communities, could identify novel bacteria involved in nutrient cycling which has remained a persistent challenge in the field of bacterial ecology (*Preheim et al., 2016*). Finally, when combined with microfluidics, direct lysis of bacteria in an emulsion, as shown here, could be used to develop or simplify single-cell genome sequencing or single-cell RNA-seq for bacteria.

## Materials and methods

### Optimizing OIL-PCR buffer for phusion and RL lysozyme compatibility

#### SYBR-Based qPCR assay for phusion polymerase activity with lysozyme

SYBR-based qPCR were set up in duplicate as follows: 25 µl reactions with 20 U/ml Phusion Hot Start Flex DNA polymerase (NEB M0535L), 1× HF Buffer, 200 µM dNTP mix (NEB N0447L), 400 nM of 519F and 786R, 1× SYBR Green (Thermo Fisher S7563), 1× ROX reference dye (Thermo Fisher 12223012), 0.5 mg/ml of BSA (NEB B9000S) when included, 0.01% Triton-X 100, and 1 µl of template DNA. Reactions were prepared with 0, 2.5, 5, 10, 20, 30, 40, and 50 U/µl of RL Lysozyme (Lucigen R1810M). qPCR was performed on the Thermo Fisher Quant Studio 3 Real-Time PCR machine with the following parameters: 98°C for 1 min, then 50 cycles of 98°C for 5 s, 54°C for 30 s, and 72°C for 30 s. Proper amplification was confirmed using melt curves: 98°C for 5 s, cool to 60°C at 1.6°C/s, and then heat to 95°C at 0.15°C/s. Ct values and melt curves were generated with the Quant Studio software V1.4 using the default software settings.

#### Lysozyme activity assay in OIL-PCR master mix

Lysozyme testing was performed in a lysozyme test buffer made from the OIL-PCR master mix with dNTPs, primers, and Phusion polymerase replaced with water and 100% glycerol (48 µl/ml). Log-phase cultures of *B. subtilis* were standardized to an $OD_{600}$ of 2 and suspended in 1× Lysozyme test buffer. Separately, lysozyme was suspended in 1× Lysozyme test buffer at 2× concentration. One hundred microliter of the lysozyme mix was aliquoted into a 96-well, clear, flat-bottomed microtiter plate before adding 100 µl of suspended culture. Lysis was monitored using a Spectramax M3 plate reader (Molecular Devices), heated to 37°C, and $OD_{600}$ measured every minute for an hour.

#### Optimizing OIL-PCR efficiency in an emulsion

Fifty microliter PCR were prepared in 1.5 ml tubes as describe in the Tube-Based OIL-PCR method below with varying concentrations of Phusion polymerase, BSA, RL lysozyme, DTT, $MgCl_2$, dNTPs, and ammonium sulfate. *E. coli* genomic DNA was used as template and amplified with universal 16S primers 519F and 796R. Reactions were emulsified 25 Hz for 30 s before aliquoting to PCR tubes and thermocycling. Final PCR products were separated from the emulsion as describe below and amplification efficiency was assessed quantitatively by SYBR-based qPCR or qualitatively by gel image band intensity.

#### Emulsion stabilization experiments

OIL-PCR test buffer was prepared similarly to the lysis activity assays with ether NEB HF buffer or detergent-free buffer (Thermo Fisher F520L), while omitting bacterial cells or RL Lysozyme. Reactions were emulsified at 25 Hz for 30 s on a Retch Mixer Mill MM 400 with adapters 11990 and 11993 (Mobio/Qiagen). Emulsion tubes were photographed before and after thermocycling and assayed by eye for coalescence. After confirming a stable emulsion, qPCR and lysis time series experiments were repeated to confirm activity of Phusion DNA polymerase and RL Lysozyme in the DF buffer.

### OIL-PCR in tube-based format for master mix optimization

#### Fusion PCR setup

All steps were performed on ice or in a 4°C centrifuge until after emulsification. Fifty microliter PCR were prepared in a 1.5 ml microcentrifuge tube with varying experimental conditions. Two microliter of bacterial cells standardized to $10^4$ cells/µl were added to 48 µl of master mix and vortexed to evenly disperse cells before adding 300 µl of cold Droplet Generation Oil for Probes (Bio-Rad 1863005). Emulsions were formed immediately after by shaking tubes at 25 Hz for 30 s on a Retch Mixer Mill MM 400 with adapters 11990 and 11993 (Mobio/Qiagen). Next, the emulsion mix was divided into four 70 µl aliquots in a PCR strip-tube and thermocycled as follows: 37°C for 5 min, 95°C for 10 min, then 38 cycles of 95°C for 5 s, 54°C for 30 s, and 72°C for 30 s, followed by final extension 72°C for 2 min. After PCR amplification, the aliquots were briefly vortexed and pooled into a clean 1.5 ml microcentrifuge tube. To break the emulsion, 50 µl of TE and 70 µl of Perfluorooctanol (Krackeler Scientific 45-370533-25G) were added and the mixture was vortexed vigorously for 30 s.

Tubes were centrifuged at 5000 G for 1 min, and the upper aqueous phase was transferred to a new PCR strip tube and purified using AMPure XP beads as described below.

## Manual AMPure XP bead cleanup

AMPure XP beads (Beckman A63880) were added at a ratio of 0.8 µl beads per 1 µl of recovered DNA, vortexed, and incubated for 5 min for DNA binding. PCR strip tubes were transferred to a 96-well magnet (Eppendorf Magnum FLX) to pull down beads for 5 min. Supernatant was removed with a multichannel pipette and the pellet was washed twice with 100 µl of 70% EtOH before drying for 10 min at room temperature. The bead pellet was suspended in 20–50 µl of TE and incubated for 5 min to elute DNA, before returning to the magnet and transferring supernatant to fresh PCR strip tubes. Eluted DNA was either run directly on a gel for qualitative analysis of amplification or used as template in qPCR assays.

## Probe-based qPCR with TaqMan probes for cell input optimization and multiplexing

### Standardization of bacterial test strains

For all experiments, bacterial type strains *Escherichia coli* MG1655 (*Blattner et al., 1997*), *Vibrio cholerae* N16961 (*Heidelberg et al., 2000*), and *Bacillus subtilis* 168 (*Kunst et al., 1997*) were inoculated from frozen glycerol stocks into 5 ml LB and grown at 37°C overnight. Cultures were diluted 1:100 in 5 ml fresh LB the next day and grown to $OD_{600}$0.4–0.8. CFU/µl at $OD_{600}$ was quantified by serial dilution of cells in LB, plating, and colony counting. Count results were used to standardize cell cultures to a stock concentration of $10^6$ CFU/µl to be diluted and used as input for OIL-PCR.

### Optimizing cell input concentration

Cultures of WT *V. cholerae* N16961 (*Heidelberg et al., 2000*) and *E. coli* MG1655 (*Blattner et al., 1997*) carrying plasmid pBAD33 (*Guzman et al., 1995*) with *cmR* were standardized to $10^4$, $10^5$, and $10^6$ CFU/µl in LB. The two strains were mixed 1:1 at each of the three concentrations, and 2 µl of cells was used as template in 50 µl tube-based OIL-PCR with fusion primers targeting the *cmR* gene. Reactions with each strain emulsified individually were run as controls. Droplet coalescence was assayed by mixing individual control reactions after emulsification, thereby ensuring the two strains were not encapsulated together. Reactions were thermocycled and recovered DNA was used as template in the probe-based qPCR to quantify specific vs non-specific fusion products.

### Probe-based qPCR assay

Probes were designed to target unique regions of *E. coli* and *V. cholerae* 16S ribosomal rRNA gene. Both probes hybridized to the antisense strand and can only be cleaved when the polymerase extended from the nested *cmR* primer, across the fusion junction, and into the 16S gene, thus distinguishing actual fused PCR products from stray fragments of 16S DNA. Probes were verified to only target their specified strain, with the *V. cholerae* probe having a VIC/NFQ MGB reporter probe and *E. coli* a FAM/NFG MGB probe. Twenty microliter qPCR were prepared in duplicate as follows: 1× Luna Universal Probe qPCR Master Mix (NEB M3004L), 300 nM of forward and reverse primer, 3.2 µM of forward and reverse blocking primers, 200 nM of *E. coli* and *V. cholerae* TaqMan probes, and 2–5 µl of recovered OIL-PCR amplicons. Reactions were amplified under the following conditions: 95°C for 1 min, then 50 cycles of 95°C for 20 s, 55°C for 20 s, and 60°C for 20 s. For analysis, Ct values were subtracted from the total number of cycles for easier interpretation.

### Primer multiplexing validation

Four strains of bacteria were used to test primer multiplexing: *V. cholerae* N16961 (*Kunst et al., 1997*) carrying *ampR* on RP4 plasmid (*Klümper et al., 2015*), WT *V. cholerae* N16961, *E. coli* 0006 (CDC and FDA Antibiotic Resistance Isolate Bank) carrying *bla*CTX-M-15, and WT *E. coli* MG1655 (*Blattner et al., 1997*), all mixed at a ratio of 1:49:10:40 with a final concentration of $10^4$ cells/µl. This mix of cells resulted in 10% of the consortium carrying *bla*CTX-M-15 and 1% carrying *amr*R to provide a more realistic depiction of the abundances of ARGs in natural stool communities. OIL-PCR was performed in a plate-based format with forward and fusion primers for both *amr*R and *bla*CTX-M-15. Each strain was tested individually as controls. Purified fusion products were assayed for correct

fusions using the probe-based qPCR assay with nested primers targeting *amrR* or *bla*CTX-M-15 in parallel reactions.

## Final OIL-PCR parameters

### OIL-PCR

The final, optimized OIL-PCR master mix is as follows: 100 U/µl Phusion Hot Start Flex DNA Polymerase, 1× DF Buffer (Thermo Fisher F520L), 250 µM dNTPs (NEB N0447L), 2 µM universal 16S reverse primer 786R, 1 µM of each target specific forward primer, 0.01 µM of each target specific fusion primer with universal 519F' tail, 1.5 mM additional MgCl$_2$, 5 mM ammonium sulfate, 5 mM DTT, 4 mg/ml BSA (NEB B9000S), 300 U/µl RL Lysozyme, 400 cells/µl Nycodenz-purified cells. Three hundred microliter emulsion oil (BioRad 1863005) was added to 50 µl reactions when performed in individual tubes, or 200 µl of emulsion oil was added to 100 µl OIL-PCR when performed in the 96-well plate format. Tubes were emulsified at 25 Hz for 30 s, while plates were sealed with a 50 µm aluminum seal (Axygen PCR-AS-600) and emulsified for two rounds of 27.5 Hz for 20 s, flipping the plate in between for consistent emulsion across rows.

The lysis and amplification program is as follows: 37°C for 5 min, 95°C for 10 min, then 38 cycles of 95°C for 5 s, 54°C for 30 s, 72°C for 30 s, before final extension of 72°C for 2 min.

dsDNase treatment and heat inactivation in OIL-PCR dsDNase treatment was not used in the initial OIL-PCR optimization or spike-in experiments. Cells were standardized to 10$^4$ cells/µl in 100 µl of PBS. One microliter of stock dsDNase (Thermo Fisher EN0771) was added to the tube and incubated at room temp for 10 min before returning to ice. Treated cells were used directly in OIL-PCR. The enzyme was inactivated immediately after emulsification (optional) by incubating 10 min in a water bath set exactly to 50°C with gentle mixing by hand every 2 min.

### Nested PCR

SYBR-based qPCR was performed on purified fusion PCR products to minimize the number of cycles for each reaction with the goal of reducing chimera formation. Amplification was performed in 20 µl reactions using the Luna Universal qPCR Master Mix (NEB M3003L) with 1× PCR master mix, 300 nM forward and reverse primers, and 2–5 µl of purified template. For multiplexed experiments, separate reactions were prepared, with one set of nested primers for each gene assayed. The following thermocycling conditions were used: 95°C for 2 min, 40 cycles of 95°C for 15 s, 55°C for 15 s, 68°C for 20 s, followed by a final extension phase at 68°C for 1 min. Melt curves were measured by heating to 95°C at 0.15°C/s. Blocking primers were not included in SYBR-based qPCR because of the strong signal from self-hybridization. Ct values were used to select the cycle number for nested amplification that was equal to the Ct value ± two cycles. Reactions that did not amplify in the qPCR were amplified with the highest number of cycles for that preparation.

Using the qPCR results to select the cycle number, nested PCR were prepared in duplicate 20 µl reactions as follows: 20 U/ml Phusion DNA polymerase, 1× HF Buffer, 2 µM dNTPs, 300 nM target gene-specific forward primer and universal reverse primer, 32 µM of each blocking primer, and 2–5 µl of template. Thermocycling was performed with variable number of cycles based on the qPCR as follows: 98°C for 3 min, then variable cycles of 98°C for 5 s, 55°C for 30 s, and 72°C for 30 s, followed by final extension 72°C for 5 min. Duplicate PCR were pooled and purified using automated AMPure XP cleanup.

### Illumina indexing PCR and library preparation

Custom indexing primers were designed based on *Spencer et al., 2015*. A set of unique, 9 bp barcodes was generated using Barcode Generator V2.8 (*Comai and Howell, 2012*). The primers are compatible with the Illumina Truseq primers and the index can be read with 8 bp instead of 9 to make them compatible with other libraries.

Indexing PCR was performed with 25 µl reactions as follows: 20 U/ml Phusion DNA polymerase, 1x HF Buffer, 2 µM dNTPs, 100 nM of unique forward and reverse indexing primers, and 2 µl of purified nested PCR template. Cycling was performed as follows: 98°C for 1 min, then 20 cycles of 98°C for 15 s, 56°C for 30 s, and 72°C for 45 s, followed by final extension 72°C for 2 min. PCR were purified using automated AMPure XP cleanup.

Indexed PCR libraries were quantified using QUANT-IT pico green dsDNA assay kit (Invitrogen P7589) and measured on the Spectramax M3 plate reader. Wells were pooled based on the measured concentration using the Eppendorf epMotion 5075vtc robot and the final pool quantified using the Qubit Broad Range Assay Kit (Thermo Fisher Q32853). Pools were run on a gel to confirm clean DNA before sequencing with MiSeq 2x250 V2 chemistry.

## Plate-based OIL-PCR with robotic automation

### Reaction setup

Ninety-six microliters of the final OIL-PCR master mix was aliquoted into a 500 µl deep-well plate (Eppendorf 00.0 501.101). Nycodenz-purified stool cells were diluted in PBS to $10^4$ cells/µl in an eight-well PCR strip for multichannel pipetting. Four microliters of cells was quickly added to the reactions with a 10 µl eight-channel pipette before sealing with an extra-thick foil seal (Axygen PCR-AS-600) and vortexed to mix. The reactions were briefly centrifuged to return liquid to the bottom of the plate, and then placed on an orbital microplate shaker (VWR 12620–926) at 1200 rpm for 30 s to further mix the cells while keeping the mix at the bottom of the wells. After mixing, the foil seal was carefully removed and 200 µl of cold emulsion oil was added using a multichannel pipette. The plate was then sealed with a fresh foil seal and shaken at 27.5 Hz for 20 s on the Retch shaker MM 400 with plate adapter (#11990). The plate was removed and turned over to shake an additional 20 s providing an even emulsion across the plate. After emulsifying, each reaction was aliquoted into four wells of a PCR plate (Eppendorf 0030 128.648) using the robot for consistency. The plates were sealed and run on the OIL-PCR fusion program described earlier.

### DNA recovery from emulsion

After amplification, the robot was used to purify the OIL-PCR Products. In short, replicate reactions were pooled into a fresh 500 µl deep-well plate, and 60 µl of TE and 70 µl of Perfluorooctanol (Krackeler Scientific 45-370533-25G) were added to each well. The plate was sealed and shaken on the Retch at 30 Hz for 40 s to thoroughly disrupt the emulsion. The plate was then centrifuged in a swing bucket rotor at 5000 Gs for 1 min to separate the phases and returned to the robot. Eighty microliters of the upper phase was aspirated from a defined height into a fresh 500 µl deep-well plate for automated Ampure XP bead purification.

### Automated AMPure XP bead purification

Eighty-five microliters of AMPure XP beads (Beckman A63880) was added to the deep-well plate containing the recovered OIL-PCR fusion products. The reactions were mixed at 1200 rpm for 1 min and incubated for 2 min for DNA binding, before transferring to the magnet (Eppendorf Magnum FLX) for 3 min. After pulldown, the supernatant was discarded and the wells were washed twice with 200 µl of 70% EtOH. After discarding the second wash, the plate was removed from the magnet and dried at room temp for 10 min before adding 50 µl of TE buffer. The plate was shaken at 1200 rpm for 1 min and incubated 2 min to elute the DNA. Finally, the plate was returned to the magnet and for 2 min and 48 µl of purified DNA was transferred to a fresh 96-well PCR plate (Eppendorf 0030 128.648).

## OIL-PCR on natural stool communities

### Nycodenz purification of stool cells

All steps were performed on ice and in a 4°C refrigerated centrifuge unless otherwise noted. Stool samples were collected in PBS + 20% glycerol + 0.1% L-cysteine and frozen at −80°C until processed. Frozen samples were thawed completely and thoroughly homogenized via vortexing. Samples were diluted at least 1:1 in cold PBS to reduce the sample viscosity and glycerol concentration as viscous samples did not to separate well with the Nycodenz. Samples were vortexed at maximum speed for 5 min to release cells from stool particles. Three hundred microliters cold 80% Nycodenz (VWR 100356–726) was aliquoted to the bottom of 2 ml microcentrifuge tubes, and 1.6 ml of stool slurry was overlaid on top without mixing the two phases. Tubes were centrifuged at 10,000 G for 40 min in a swing bucket rotor to separate cells. After centrifugation, the upper phase was removed with a pipetted and 500 µl of cold PBS was used to wash the bacterial cell pellet from the insoluble stool fraction. The suspended cells were removed, and the pellet was washed a second time with

500 µl of PBS. Cells were centrifuged at 50 g for 1 min to pellet any large particles that carried over from the Nycodenz purification and the upper phase was passed through a 40 µm nylon mesh screen (Falcon 352235) to remove any residual stool debris or large cell clumps. Samples of each preparation were diluted 1:1 PBS + 20% glycerol for whole cell storage. Lastly, purified cells were diluted and imaged at 100× magnification within a 20 µm counting chamber (VWR 15170–048). Images were analyzed using FIJI/ImageJ 1.52 p (Java 1.8.0_172) to manually count cells and calculate cell concentration in glycerol stocks.

### Spike-in experiment

This experiment was performed using the individual tube-based format of OIL-PCR. Nycodenz-purified stool and *E. coli* carrying pBAD33 (*Guzman et al., 1995*) with *cmR* was standardized to $10^4$ cells/µl. *E. coli* cells were mixed with the stool samples at a ratio of 1:10, 1:100, and 1:1000, and the mixed cultures were added to OIL-PCR containing the *cmR* primer set. Reactions were emulsified, lysed, and thermocycled, and fusion products were purified manually. Nested PCR was performed with the nested *cmR* primer before indexing, pooling, and sequencing.

### Lysozyme, dsDNase, heat experiment

Nycodenz-purified human and chicken stool cells were standardized to $10^5$ cells/µl and incubated with or without dsDNase at room temperature for 10 min. OIL-PCR master mix was prepared with and without Lysozyme using universal 16S rRNA primers i519F and i786R. Cells were added to the OIL-PCR and emulsified. Emulsions were either incubated at 50°C or room temperature for 10 min before aliquoting to PCR plates and running the OIL-PCR fusion program. Amplicons were purified, indexed, and submitted for Illumina sequencing as described above.

## OIL-PCR for detection of *bla* genes in neutropenic patients

### Sample collection and metagenomic assemblies

Samples were collected and sequenced, and metagenomic assemblies were prepared as described in *Kent et al., 2020*. Briefly, informed consent and consent to publish were obtained from individuals receiving a hematopoietic stem cell transplant at NewYork-Presbyterian Hospital/Weill Cornell Medical Center. Serial stool samples were obtained from consenting patients. Consent documents and procedures were approved by the Institutional Review Boards at Weill Cornell Medical College (#1504016114) and Cornell University (#1609006586). Samples were either frozen 'as is' (for metagenomic sequencing) or homogenized in phosphate-buffered saline (PBS) + 20% glycerol before freezing (for OIL-PCR). DNA was isolated from samples destined for metagenomic sequencing using the PowerSoil DNA Isolation Kit (Qiagen) with additional proteinase K treatment and freeze/thaw cycles recommended by the manufacturer for difficult-to-lyse cells. Extractions were further purified using 1.8 volumes of Agencourt AMPure XP bead solution (Beckman Coulter). DNA was diluted to 0.2 ng/µl in nuclease-free water and processed for sequencing using the Nextera XT DNA Library Prep Kit (Illumina).

### Design and validation of OIL-PCR fusion primers

ARG variants for the three *bla* genes were downloaded from the CARD database (*Alcock et al., 2020*) and aligned in Snapgene using default MUSCLE parameters. Conserved regions were identified manually, and degenerate primers were designed to capture as many variants of the genes as possible. Primers were selected for GC content between 40 and 60% and an annealing temperature of 58°C based on the Snapgene calculation. Degenerate bases were limited to three per primer and no less than 5 bp from the 5' end.

Strains acquired through the CDC and FDA Antibiotic Resistance Isolate Bank carrying multiple variants of each gene (*Supplementary file 1*) were used as template for testing *bla* primers. At least three sets of primers were designed and tested in every possible combination using the OIL-PCR master mix without emulsion to find a set of three primers that provided clean fusion amplification. Lastly, working primer sets were tested in an emulsion on whole cells to confirm amplification in OIL-PCR.

Fusion primers targeting Tn3 transposon genes were designed using scaffolds from the metagenomic assemblies and tested on *Klebsiella* isolate DNA from patient B335.

## OIL-PCR on neutropenic patients

All OIL-PCR were performed with the plate-based protocol including dsDNase treatment and heat inactivation. Whole bacterial cells were purified with Nycodenz, quantified, and standardized to $10^4$ cells/µl in PBS before treating with dsDNase. For multiplexed experiments, reactions were prepared in quadruplicate with three sets of primers targeting the three *bla* genes in each reaction. The single-plex reactions were prepared in triplicate with only one primer set per reaction. In all cases, the reactions followed the standard plate-based protocol with automation, including heat inactivation of the dsDNase after emulsification. Nested PCR, indexing, and library preparation were performed as described above.

## Romboutsia-specific OIL-PCR

CRIB primers (*Gerritsen et al., 2014*) were modified to form a fusion product with all three *bla* genes; however, only the $bla_{TEM}$ primer set amplified when tested. Using only the $bla_{TEM}$ primer set, OIL-PCR was performed with dsDNase treatment, in triplicate, using the plate-based format with automation. Nested PCR, indexing, and library preparation were performed as described above.

## Computational methods

### Processing 16S rRNA sequencing

Raw reads were merged using usearch (*Edgar, 2010*) (V 11.0.667) -fastq_mergepairs (maxdiffs: 20, pctid: 85, minmergelen: 283, maxmergelen: 293) before trimming primers and quality filtering with usearch -fastq_filter (maxee: 1.0). Unique reads were filtered using usearch -fastx_uniques, and OTUs were clustered based on 97% identity with usearch `-cluster_otus`. OTU tables were generated with usearch -otutab, and taxonomy was assigned with RDP classifier implemented in MOTHUR classify.sequs (1.38.1) against silva v132. Rarefaction curves were generated using QIIME1 (*Caporaso et al., 2010*) (v1.9) multiple_rarefaction.py (-m 10, -x 100000, -s 100, -n 5, -k). Scripts have been uploaded to the GitHub repository (https://github.com/pjdiebold/OIL-PCR_Linking_plasmid-based_beta-lactamases; *Diebold, 2021*; copy archived at swh:1:rev:6d5b25dfa6d67703f06f74c24f7efe27bcf9d8dd).

### Processing OIL-PCR sequencing

Raw reads were merged using usearch (*Edgar, 2010*) (V 11.0.667) -fastq_mergepairs (maxdiffs: 10% of expected overlap, pctid: 85, minmergelen: expected length-15, maxmergelen: expected length +15) before trimming primers and quality filtering with usearch -fastq_filter (maxee: 1.0). Unique reads were filtered using usearch -fastx_uniques. Reads were split at the fusion junction into 16S and target reads using cutadapt V2.1 (*Martin, 2011*) because of its tolerance for PCR errors, which are often introduced in the fusion junction of the OIL-PCR amplicons. The 16S reads were clustered based on 97% identity with usearch -cluster_otus, OTU tables were generated with usearch -otutab, and taxonomy was assigned with RDP classify implemented in mothur (*Schloss et al., 2009*) classify.sequs (1.38.1) against SILVA (*Quast et al., 2013*) v132. Target reads were identified by blasting against a custom database of expected sequences with blastn (*Camacho et al., 2009*) (v2.9.0). 16S taxonomy and target read identity were then reassociated using a custom python script to parse the files. Detections were defined by taxa – target associations that make up 0.5% of the total reads across replicates. Scripts have been uploaded to the GitHub repository (https://github.com/pjdiebold/OIL-PCR_Linking_plasmid-based_beta-lactamases; *Diebold, 2021*).

### Strain-level analysis of *Romboutsia* in metagenomes

Metagenomic reads from each time point were aligned to the *R. timonensis* reference genome (Refseq accession code: GCF_900106845.1) using BWA mem (v0.7.17, -a) (*Li, 2013*). Reads aligning to the reference genome were then assembled using SPAdes (v3.14.) (*Nurk et al., 2013*). To determine the presence and identity of strains from each time point, AMPHORA (*Wu and Scott, 2012*) (v2, marker identification step only) was used to identify the sequences of 30 marker genes within each assembled *R. timonensis* genome. The marker genes identified by AMPHORA were then mapped (Diamond, v2.0.4) (*Buchfink et al., 2015*) to the BLAST (*Altschul et al., 1990*) nr database for taxonomic annotation (BLAST nr database downloaded 2018). DNA sequences of the marker genes that mapped to *R. timonensis* were retained for further analysis. Genes from time point 2 and

time point 3 were aligned to one another (BLAST blastn, v2.9.0) (*Camacho et al., 2009*), and then sequences from time point 1 were aligned against sequences of the same gene from time point 2, once the sequences at time 2 and time 3 were determined to be the same. To determine how abundant each marker gene, and all of its variants, are at each time point, metagenome reads from each time point were mapped to its own set of marker gene sequences (BWA mem, v0.7.17, -a) (*Li, 2013*). Read counts were normalized for the length of each gene and the total number of reads sequenced per sample (RPKM) (*Mortazavi et al., 2008*).

## Acknowledgements

We would like to thank the following individuals and organizations for their generosity in providing us with strains and plasmids: Tobias Dörr (STRAINS), Barth Smets (RP4), John Helmann (*B. subtilis*), and the CDC and FDA Antibiotic Resistance (AR) Isolate Bank. We would like to thank Sarah Spencer for technical advice. This study was funded by the Centers for Disease Control (OADS BAA 2016-N-17812) and by the National Sciences Foundation (Awards #1661338 and #1650122). FN is a SUNY Diversity Fellow. MJS is funded by the NIAID (K23 AI114994). ILB is funded by the NIH (1DP2HL141007-01) and is a Sloan Foundation Research Fellow, a Packard Fellowship in Science and Engineering, and a Pew Foundation Biomedical Scholar.

## Additional information

### Funding

| Funder | Grant reference number | Author |
| --- | --- | --- |
| Centers for Disease Control and Prevention | OADS BAA 2016-N-17812 | Michael J Satlin<br>Ilana Brito |
| National Science Foundation | 1661338 | Ilana Brito |
| National Science Foundation | 1650122 | Ilana Brito |
| National Institutes of Health | K23 AI114994 | Michael J Satlin |
| National Institutes of Health | 1DP2HL141007-01 | Ilana Brito |
| Pew Charitable Trusts | | Ilana Brito |
| Alfred P. Sloan Foundation | | Ilana Brito |
| David and Lucile Packard Foundation | | Ilana Brito |
| State University of New York | | Felicia N New |

The funders had no role in study design, data collection and interpretation, or the decision to submit the work for publication.

### Author contributions

Peter J Diebold, Conceptualization, Data curation, Software, Formal analysis, Validation, Investigation, Visualization, Methodology, Writing - original draft, Writing - review and editing; Felicia N New, Software, Formal analysis, Visualization, Methodology, Writing - review and editing; Michael Hovan, Resources, Data curation; Michael J Satlin, Conceptualization, Supervision, Funding acquisition, Writing - review and editing; Ilana L Brito, Conceptualization, Resources, Formal analysis, Supervision, Funding acquisition, Investigation, Methodology, Writing - original draft, Project administration, Writing - review and editing

### Author ORCIDs

Peter J Diebold https://orcid.org/0000-0002-9189-9941
Ilana L Brito https://orcid.org/0000-0002-2250-3480

## Ethics

Human subjects: All human subjects research was approved by the Weill Cornell Medicine IRB (#1504016114) and Cornell University IRB (#1609006586). Informed consent and consent to publish were obtained from individuals receiving a hematopoietic stem cell transplant at NewYork-Presbyterian Hospital/Weill Cornell Medical Center. Serial stool samples were obtained from consenting patients. Consent documents and procedures were approved by the Institutional Review Boards at Weill Cornell Medical College (#1504016114) and Cornell University (#1609006586).

## Decision letter and Author response

Decision letter https://doi.org/10.7554/eLife.66834.sa1
Author response https://doi.org/10.7554/eLife.66834.sa2

## Additional files

### Supplementary files

• Supplementary file 1. Strains and plasmids used in this study.

• Supplementary file 2. Primers used in this study.

• Supplementary file 3. OIL-PCR primer set table.

• Transparent reporting form

### Data availability

All fusion PCR amplicons are deposited in SRA under the accession number PRJNA701446. The metagenomic samples analyzed can be obtained through SRA with under the accession number PRJNA649316.

The following dataset was generated:

| Author(s) | Year | Dataset title | Dataset URL | Database and Identifier |
|---|---|---|---|---|
| Diebold PJ, Brito IL | 2021 | Linking plasmid-based beta-lactamases to their bacterial hosts using single-cell fusion PCR | https://www.ncbi.nlm.nih.gov/bioproject/PRJNA701446 | NCBI BioProject, PRJNA701446 |

The following previously published dataset was used:

| Author(s) | Year | Dataset title | Dataset URL | Database and Identifier |
|---|---|---|---|---|
| Kent A, Vill A, Brito IL | 2020 | Hi-C and metagenomic sequencing of neutropenic patient microbiomes | https://www.ncbi.nlm.nih.gov/bioproject/PRJNA649316/ | NCBI BioProject, PRJNA649316 |

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

## Appendix 1

### Full lab protocol for OIL PCR

OIL-PCR protocol

*Please email Peter Diebold, pd378@cornell.edu, for questions regarding this protocol

## General workflow

1. Design and test primers
2. Nycodenz purify cells from sample (day 1)
3. Quantify cell concentration of purified cells (day 1)
4. Perform OIL-PCR (day 2)
5. Break the emulsion and recover the aqueous phase (day 2)
6. Bead purify the fusion PCR products from the aqueous phase (day 2)
7. Perform nested qPCR to choose cycle number for each sample/primer combination (day 3)
8. Perform the nested PCR with cycle number from qPCR (day 3)
9. Bead purify the DNA (day 3)
10. Perform the Illumina Indexing PCR (day 4)
11. Bead Purify the DNA (day 4)
12. Quantify the DNA concentration (day 4/5)
13. Pool reactions for Illumina sequencing (day 5)

## Design primers

1. Collect sequences for the desired target. For ARGs, I pulled all the available sequence from the CARD database.
   a. In many cases the diversity of the genes is too great to design a single primer set. In this case I looked through the genes on CARD and annotated the ones that were most prevalent or medically relevant and tried to design primers for them.
   b. Sometimes more than one primer set will be needed to target a diverse group of genes.
2. Identify potential priming regions
   a. First, I mark regions with GC content between 40 and 60%. Snapgene has a function to display GC content as a graph.
   b. I also align the sequence variants and mark regions that are highly similar for priming.
   c. It is also advantageous to design primers which span a region of dissimilarity for detecting gene variants when possible.
3. Design multiple potential primers with the following parameters
   a. GC content between 40 and 60%
   b. Tm in snapgene of approximately 58 (This is the number I've always used)
   c. I try to avoid too many degenerate bases, but they are often unavoidable in which case I try to keep degeneracies away from the 3' end
   d. Design multiple primers without worrying about fragment size immediately
   e. When designing highly specific primers (i.e. species targeting), the NCBI primer blast is a useful tool
4. Search for potential combinations of the primers which will work for OIL
   a. The fusion primer and Round one primer can be far apart, although I try to keep it short if possible
   b. The nested primer and the fusion primer fragment should not be more than 200 bp including the primers
   c. I will often design primers so that I can try them in multiple combinations to see which work best together.
5. Add the fusion primer and nested primer tails to the 5' end
   a. Fusion tail: GWATTACCGCGGCKGCT
   b. Nested tail: ACACGACGCTCTTCCGATCT
6. Test the primers:
   a. Primers should be tested in a mock OIL-PCR mastermix

 i. It is the same as the normal master mix, but without lysozyme and using the manufacturers recommended concentration of Phusion polymerase. I usually do 20–25 µl reactions

 b. I like to do SYBR based qPCR for the nested PCR, but it's not necessary. Just make sure something amplifies. Simply add 1× SYBR and optionally 1× ROX to the master mix

 c. I also will sequence the final fusion constructs

## Nycodenz purify cells
Reagents/equipment

- Stool stored in PBS + 20% glycerol + 0.1% L-cysteine
- Cold PBS
- Cold PBS + 20% glycerol + 0.1% L-cysteine
- Cold 80% Nycodenz (VWR 100356–726)
- 2 ml microcentrifuge tubes
- 40 µm nylon mesh screen (Falcon 352235)
- Cryogenic vials
- Dry Ice/EtOH slurry or liquid N2
- Vortexer
- Refrigerated centrifuge with a swing bucket rotor cooled to 4°C
- 1 ml filter pipette tips with approximately 1 cm cut from the end for pipetting stool (wide bore tips to not have a large enough orifice)

Protocol: Perform all steps on ice and in a refrigerated centrifuge at 4°C

1. Vortex stool sample to thoroughly homogenize
2. Dilute approximately 800 µl of stool 1:1 in cold PBS to reduce sample viscosity (dilute further for particularly viscous samples). Use the cut pipette tips to transfer stool.
   - Note: too little stool results in a thin stool pellet which makes removing the cells more difficult; however, too much stool/glycerol can interfere with the density gradient. Performing a run with 'practice' stool is recommended.
3. Add 300 µl of 80% Nycodenz solution to the bottom of a 2 ml microcentrifuge tube. Be sure to pipette the solution directly to the bottom of the tube
4. Overlay 1.6 ml of stool slurry on top of the Nycodenz. Be gentle so that the stool does not mix with the Nycodenz.
5. Centrifuge the tubes for 40 min at 10,000 G in a swing bucket rotor
6. Remove the upper phase and discard
7. Add 500 µl of cold PBS and pipette up and down to wash the lighter colored cell layer from the stool pellet. Sometimes there is a lighter and darker layer of cells with the darker being harder to suspend. Check on a scope to confirm relatively pure cell fractions.
8. Pass cells through the 40 µm nylon mesh screen
9. Optional: Filter cells through a 5–8 µm filter to remove cell clumps and clean the cells further. Expect significant loss of cells in the filter
10. Dilute cells 1:1 in PBS + 20% glycerol + 0.1% L-cysteine. Set some aside for cell quantification
11. Freeze multiple aliquots of the purified cells in cryogenic vials. Ideally flash freeze and store at −80°C

## Quantify cell concentration
Reagents/equipment

- Nycodenz-purified cell fraction
- 20 µm counting chamber (VWR 15170–048)
- Microscope with 40× phase contrast objective

Protocol

1. Dilute cells to a countable concentration. Too many cells or too few will make counting difficult. Usually 1:10 or 1:100 is good.
2. Pipette 10 µl of cells on the counting chamber and add the coverslip
3. Place the slide on the microscope and capture images of at least five squares
   - Note: The scope cannot perfectly focus on all planes within the depth of the counting chamber. You must focus somewhere in the middle so that most cells are blurry but visible for counting
4. Use FIJI to count cells in each square and calculate the final concentration based on the counting chamber volume and dilution factor
   - Because of the imperfect focus, the counting must be done by hand using the 'cell-counter' plugin

Note: Fluorescent staining and imaging the cells could improve counting accuracy, but any stray stool particles will fluoresce strongly. Also, a flow cytometer could be used if available.

## OIL-PCR
### Reagents/equipment

- Thawed Nycodenz-Purified Cells
- Cold 1× PBS
- Cold BioRad Emulsion Oil (BioRad 1863005)
- dsDNase (Thermo Fisher EN0771)
- PCR Reagents:
  - 5X DF Buffer (Thermo Fisher F520L)
  - dNTPs (NEB N0447L)
  - 100 µM 16S reverse primer AP27 (TTTTTTGCTCTTCCGATCTGGACTACHVGGGTWTCTAAT)
  - 100 µM forward primer
  - 10 µM fusion primer (5′ tail GWATTACCGCGGCKGCT)
    - Multiplexed reactions will have up to 3 forward and three fusion primers
  - MgCl$_2$ (NEB M0535L)
  - 100 mM Ammonium Sulfate
  - 100 mM DTT
  - BSA (NEB B9000S)
  - Ready-Lyse Lysozyme (Lucigen R1810M)
  - Phusion Hot Start Flex DNA Polymerase (NEB M0535L)
- 96-Well PCR plates (Eppendorf 0030 128.648)
  - Do not swap plates as it has been reported that the different plates can affect emulsion stability
- Retch Mixer Mill MM 400 with plate adapters (Qiagen/MoBio #11990).
  - Also use adapter 1193 for tube-based format
  - Any bead beater would be a suitable alternative to the Retch.
  - A vortexer could also be used however we have not verified the parameters

### Equipment for automated protocol

- 500 µl Deep-Well Plate (Eppendorf 00.0 501.101)
- Thick Foil Seal (Axygen PCR-AS-600)
- 8-Well PCR Strip Tube
- Disposable reservoir for multichannel pipettes
- 10 µl multichannel pipette
- 200 or 300 µl multichannel pipette
- Eppendorf EP Motion setup to run 'OIL_PCR-A' program

## Protocol

1. Standardize cell stock to $10^5$ cells/µl in cold PBS using the calculated concentration
   - Standardize directly to $10^4$ cell/µl if stocks are dilute
2. Add 1 µl of Ready-Lyse Lysozyme to each tube of cells and incubate for 10 mins at RT. Return cells to ice when complete
3. Prepare the PCR MasterMix:
   - 50 µl reactions for the manual tube-based format
   - When multiplexing, there will be up to three forward and three fusion primers total. Adjust the master mix to account for the extra primers
4. Aliquot 96 µl of OIL mastermix into each well of a 500 µl DWP on ice
   - Or 48 µl into a 1.5 ml tube for the tube-based method
5. Dilute the dsDNase treated cells 1:10 to a final concentration of $10^4$ cell/µl in an eight-well PCR strip tube for multichannel pipetting
   - This dilution step reduces the final concentration of dsDNase in the OIL reaction to prevent degradation
6. Using the 10 µl multichannel, transfer 4 µl of cells to OIL-PCR mastermix. Gently pipette to mix
   - Add the cells directly to the bottom of each well. Avoid getting any on the side of the plate
   - It is extremely important to keep the cells cold and work fast to prevent premature lysis of cells before emulsification
   - Add 2 µl of cells individually when performing the tube-based method
7. Seal the plate and vortex to mix
8. Quick spin the plate to return all liquid to the bottom of the plate
9. Carefully vortex the reactions a second time for 30 s. Try to keep the liquid at the bottom of the wells. A high-speed plate shaker is best if available
   - Note: Mixing the cells evenly through the master mix is extremely important. Unmixed droplets of cells on the side of the tube will result in poor isolation of cells.
10. Quickly add 200 µl of emulsion oil to each well using a multichannel pipette
    - 300 µl for the tube-based method
11. Seal the plate with a foil seal and shake for 20 s at 30 Hz
    - 25 Hz for 30 s in the tube based. Flipping is unnecessary if the tubes are in the center
12. Flip the plate so the inside arc is now on the outside and shake for another 20 s
13. After emulsification the reaction can be kept at room temperature to for Lysis to begin
14. Run the OIL_PCR-A program to consistently aliquot the emulsion to PCR plate
    - This step can be done by hand (70 µl mix to four wells of the plate), but it is difficult to evenly distribute the emulsion between wells. The robot is used to properly mix the emulsion before each transfer
    - Perform the transfer by hand with the tube-based method
15. Seal the plates and run the OIL-PCR thermocycle program:
    - Note 1: The program incubates at 30°C and not 37°C. This was implemented because of concern that the dsDNase or endogenous nucleases could degrade DNA too quickly at higher temperatures. 37 would likely work better for lysis but the method has not been changed for consistency
    - Note 2: Slow temperature ramp rates were used to allow even heating through the emulsion
    - Note 3: The emulsion will separate to the top of the reaction and congeal which is normal

| Reagent | Stock concetration | Final concentration | Volume (µl) |
|---|---|---|---|
| H$_2$O | | | to 100 µl |
| DF Buffer | 5× | 1× | 20 |
| dNTPs | 10 mM | 250 M | 2.5 |

*Continued on next page*

*continued*

| Reagent | Stock concetration | Final concentration | Volume (µl) |
|---|---|---|---|
| 16 S-R AP27 | 100 µM | 2 µM | 2 |
| pForward | 100 µM | 1 µM | 1–3 |
| pfuse | 10 µM | 0.01 µM | 0.1–0.3 |
| MgCl$_2$ | 50 mM | 1.5 mM | 3 |
| AmSulfate | 100 mM | 5 mM | 5 |
| DTT | 100 mM | 5 mM | 5 |
| BSA | 20 mg/ml | 4 mg/ml | 20 |
| Lysozyme full | variable | 300 U/µl | 0.792 |
| Polymerase | 2000 U/ml | 100 U/µl | 5 |
| template | $10^4$ cells/µl | 400 cell/µl | 4 |
| Total | | | 100 |

| | | 5:00 | 30 |
|---|---|---|---|
| | | 10:00 | 95 |
| 38× | | 0:05 | 95 |
| | | 0:30 | 54 |
| | | 0:30 | 72 |
| | | 2:00 | 72 |
| | | hold | 4 |

## Breaking the emulsion
Reagents/equipment

- Perfluorooctanol (Krackeler Scientific 45-370533-25G)
- TE

For automated version:

- Centrifuge capable of spinning deep-well plates
- 300 µl filter pipette tips for the robot
- 300 µl multichannel tool
- 30 ml reservoirs

## Automated version

1. Open the robot protocol for 'OIL_PCR_B' or 'OIL_PCR_B_Ampure'
   a. The ampure version transitions directly to the bead cleanup after breaking the emulsion
2. Setup the stage as shown in the program
3. Vortex the plate to break up the emulsion before carefully opening
4. The robot will first pool the reactions into a 500 µl 96-well deep-well plate
5. The robot will then add 60 µl TE and 70 µl perfluoro-1-octanol
6. Seal the plate with foil and place on the shaker at 30 Hz for 20 s per side
7. Spin down the plate 5000G for 1 min
8. Return the plate to the robot and it will remove the lower oil phase and discard it in the waste reservoir
9. Then it will pipette off the upper aqueous phase into a 96-well PCR plate, or the other half of the DWP if using the ampure version

 a. If doing the ampure version, it will continue as describe in the AMPure XP automated protocol

## Manual version

1. Vortex the plate to break up the emulsion before carefully removing the foil seal without cross contaminating wells
2. Pool the four reactions into a 1.5 ml tube, being sure to mix well between pipette steps to capture as much of the emulsion as possible
3. Centrifuge at 500G for 1 min and remove the lower oil phase
4. Add 50 µl TE and 70 µl Perfluorooctanol
5. Vortex at max speed for 30 s
6. Centrifuge at 500G for 1 min
7. Carefully remove the upper phase and transfer to a PCR strip for Ampure XP bead purification

## Ampure XP cleanup

### Reagents

- AMPure XP beads (Beckman A63880)
  - 96-Well plate magnetic separator (Eppendorf Magnum FLX)
- Any magnet will work for the manual protocol
- TE
- 70% EtOH

Manual specific:

- Multichannel reservoir
- 100, 200, or 300 µl multichannel pipette

Automation specific:

- 300 µl and 1000 µl filter tips
- 300 and 1000 µl multichannel tools
- 30 and 100 ml reservoirs
- 500 µl deep-well plate (Eppendorf 00.0 501.101)
- 96-Well PCR plate for elution (Eppendorf 0030 128.648)

### Manual protocol

This can be done in either a full 96-well plate or also individual 8-well strip tubes. If using strip tubes, you will need to fashion some kind of adapter to hold them upright in the magnet. The top of some 200 µl tip boxes often works well.

1. Add a ratio of 0.8x beads to each reaction (e.g., 80 µl beads for 100 µl PCR)
 a. It's better to have too much than too little
2. Pipette or vortex to mix and allow 5 min for the DNA to bind the beads
3. Perform a brief spin to return all liquid to the bottom of the wells
4. Place the tubes on the magnet for 2 min to pull down the beads
5. Use a multichannel to remove the supernatant
6. Use a multichannel to add 100 µl of EtOH to each well. Pipette up and down to wash without disturbing the beads and immediately remove and discard the supernatant
7. Repeat step six for a second wash
8. Remove from the magnet and dry at RT for 10 min
9. Add the desired amount of TE to each well (I usually default to 25 or 50 µl)
10. Mix well either by pipetting or vortexing
11. Allow five mins for the DNA to fully elute
12. Place on the magnet and allow 2 min for the pull-down
13. Transfer the supernatant to a fresh plate/strip-tubes with a multichannel pipette

Automated protocol

1. Setup the robot as described in the 'Ampure Cleanup' Protocol
   a. Fill two 30 ml reservoirs with appropriate volumes of Beads and TE
   b. Fill a 100 ml reservoir with EtOH
   c. Place the tips, reservoirs, waste, magnet, and plates as shown in the program
2. Adjust the Ampure XP transfer volume to be 0.8× of the PCR volume
3. Adjust the TE volume for the desired elution
4. Begin the program. It performs all the same steps as the manual one.

## Run nested qPCR
Reagents

- Luna universal qPCR master mix (NEB M3003L)
- Nested Target Primers
- Reverse 16S Primer AP28

Protocol

For multiplexed reactions, there will be an individual qPCR assay for each of the genes. DO NOT MULTIPLEX THE NESTED PCR REACTIONS
   1. Make the qPCR master mix with the following recipe

| Reagent | Stock concetration | Final concentration | Volume (µl) |
| --- | --- | --- | --- |
| $H_2O$ | | | to 20 |
| Luna Buffer | 2× | 1× | 10 |
| Nest Primer | 100 uM | 300 nM | 0.06 |
| 16 S-R AP28 | 100 uM | 300 nM | 0.06 |
| Template | | | 2–5 |

   2. Aliquot the master mix into a qPCR plate and use a multichannel pipette to add template to the reaction
   3. Run the reactions with the following program

| | | **2:00** | **95** |
| --- | --- | --- | --- |
| 38× | | 0:15 | 95 |
| | | 0:15 | 55 |
| | | 0:20 | 68 |
| | | 1:00 | 65 |
| | | 0.15°C /s | 95 |

4. Check melt curves to confirm clean amplification
5. Select cycle numbers for each sample equal to the Ct value ± two cycles

## Run nested PCR
Reagents

- 5X HF Buffer (NEB M0535L)
- dNTPs (NEB N0447L)
- 100 µM 16S reverse primer AP28 (GAGTTCAGACGTGTGCTCTTCCGATCTGGACTAC)

- 100 µM Nested primer (5' Tail ACACGACGCTCTTCCGATCT)
- 100 µM Blocking F (TTTTTTTTTTCAGCMGCCGCGGTAATWC/3SpC3/)
- 100 µM Blocking R (TTTTTTTTTTGWATTACCGCGGCKGCTG/3SpC3/)
- Phusion Hot Start Flex DNA Polymerase (NEB M0535L)
- OIL-PCR Template

## Protocol

For multiplexed reactions, there will be an individual reaction for each of the genes. DO NOT MULTIPLEX THE NESTED PCR REACTIONS!

1. Prepare enough mastermix without template for all samples as follows:

| Reagent | Stock concetration | Final concentration | Volume (µl) |
|---|---|---|---|
| H2O | | | to 100 µl |
| HF Buffer | 5× | 1× | 6 |
| dNTPs | 10 mM | 200 µM | 0.6 |
| Nest Primer | 100 µM | 30 nM | 0.09 |
| 16 S-R AP28 | 100 µM | 30 nM | 0.09 |
| Block F | 100 µM | 3.2 µM | 0.96 |
| Block R | 500 µM | 3.2 µM | 0.96 |
| Polymerase | 2000 U/µl | 20 U/µl | 0.3 |
| template | | | 2–5 |
| Total | | | 30 |

2. Aliquot master mix minus template volume to wells of a 96-well plate
3. Add Purified template with a multichannel pipette
4. Mix the reactions and transfer 15 µl of each reaction to a fresh plate for replicates (2 × 15)
5. Run the reactions as follows:

| | 2:00 | 98 |
|---|---|---|
| Variable | 0:05 | 98 |
| | 0:30 | 55 |
| | 0:30 | 72 |
| | 5:00 | 72 |

6. After cycling, pool the replicates and perform the Bead cleanup

a. Thermolabile Exonuclease I (NEB M0568S) could be used instead of a bead cleanup at this step to save time and reagents. The exonuclease will degrade the primers from the nested reaction and then is head inactivated.

## Run index PCR

### Reagents

- 5X HF Buffer (NEB M0535L)
- dNTPs (NEB N0447L)
- 5 µM Forward Index
- 5 µM Reverse Index
  - Alternatively, a plate of premixed primers can save time in the long run. In other words, prepare a 96-well plate of primers, where each well has a unique combination of index primers
  - Primer sequences are in the *Supplementary file 2*
- Phusion Hot Start Flex DNA Polymerase (NEB M0535L)
- Nested PCR Template

Prepare enough mastermix for all samples, without template or primer as follows:

| Reagent | Stock concetration | Final concentration | Volume (µl) |
|---|---|---|---|
| H$_2$O | | | to 25 µl |
| HF Buffer | 5× | 1× | 5 |
| dNTPs | 10 mM | 200 µM | 0.5 |
| F Index | 5 µM | 100 nM | 0.5 |
| R Index | 5 µM | 100 nM | 0.5 |
| Polymerase | 2000 U/µl | 20 U/µl | 0.25 |
| template | | | 2–5 |
| Total | | | 25 |

8. Aliquot the master mix to a 96-well plate (minus template and primer volume)
9. Add index primers to the plate individually
10. Primers can be aliquoted into PCR strip tubes for multichannel pipetting across the plate
a. dilution of primers to 1 uM can make pipetting easier
b. Use a multichannel to transfer purified, nested PCR template to each well
11. Run the following program for indexing:
12. After amplification, perform a bead purification of the reactions

| | | **1:00** | **98** |
|---|---|---|---|
| 20 cycles | | 0:15 | 98 |
| | | 0:30 | 56 |
| | | 0:45 | 72 |
| | | 2:00 | 72 |

## Quantify the DNA concentration

Use the QUANT-IT pico green dsDNA assay kit (Invitrogen P7589) as described in the manufactures instructions to quantify the concentration of DNA and measure using a fluorometric plate reader.

## Pool the reactions

Reactions were pooled to a standard concentration from all reactions and submitted for sequencing.

