## [Decision Letter]

**Acceptance summary:**

Reviewers particularly praised the development of an accessible high-throughput technique as an alternative to HiC or single cell whole genome sequencing to associate mobile antibiotic resistance genes with their bacterial hosts in complex microbial populations. This is an improvement of the previously published epicPCR and relies on cellular emulsion and fusion PCR in one step. Tracking horizontal gene transfer of mobile resistance genes is exceptionally important in the context of AMR and its burden on public health. In addition, the method can be applied to other studies on complex microbial communities beyond antibiotic resistance.

**Decision letter after peer review:**

Thank you for submitting your article "Linking plasmid-based beta-lactamases to their bacterial hosts using single-cell fusion PCR" for consideration by *eLife*. Your article has been reviewed by 3 peer reviewers, and the evaluation has been overseen by a Reviewing Editor and Dominique Soldati-Favre as the Senior Editor. The following individual involved in review of your submission has agreed to reveal their identity: Martial Marbouty (Reviewer #2).

Reviewers identified a few points that need addressing to strengthen the study further.

Essential revisions:

1) Please show the feasibility of discrimination between physical association of the bacteria or transfer of resistance genes in a defined mix of bacteria in vitro, for example by using primers specific to non-transferable genes as you suggested in discussion.

2) Provide an additional figure with the global computational pipeline used so that readers better understand how the results were obtained.

3) Provide a detailed protocol with a list of reagents and equipment.

4) Please provide a clearer comparison of your results with the HiC linkage obtained in Kent et al., rather than just the metagenomic data as it would strengthen the claims and results.

5) you compare the results of sequencing with a custom database of expected sequences but what are the results if compared to the NCBI database?

6) The analysis of the effect of lysozyme on improving capture of gram positive bacteria is restrained to phylum. A more in-depth analysis (at the class level for instance) would be useful. For instance, it exists a clear difference between chicken and human stool samples with or without lysozyme. Why? Is it linked to some specific phylum?

---

## [Author Response]

Essential revisions:1) Please show the feasibility of discrimination between physical association of the bacteria or transfer of resistance genes in a defined mix of bacteria in vitro, for example by using primers specific to non-transferable genes as you suggested in discussion.

Thank you to all reviewers for this comment—we agree that this experiment strengthened the results in the manuscript. At the reviewers’ suggestion, we designed OIL-PCR fusion primers targeting two *Klebsiella pneumoniae* housekeeping genes (*rpoB* and *glmS*) and two *Romboutsia timonensis* genes (*rpoB* and *nusA*). While these genes are universally present, they are divergent enough between strains that primers can be designed specifically at the species level. Using these primer sets, along with CTX-M primers as a control, we ran OIL-PCR to see if we found *Klebsiella* marker genes fused to *Romboutsia* 16S sequences or vice versa. *Klebsiella* primers were multiplexed with CTX-M and the *Romboutsia* primers were assayed in separate reactions to rule out the possibility of PCR chimeras during library preparation.

The results showed that *Romboutsia* 16S sequences were indeed fused to both *Klebsiella* marker genes, mirroring the same pattern as was seen for the ARGs and transposase genes assayed. The CTX-M control also presented the same pattern as previously demonstrated. Although the *nusA* primers failed to amplify, the *RomboutsiarpoB* sequences were correctly fused to *Romboutsia* 16S. *Klebsiella* 16S was also found fused to *RomboutsiarpoB*, however, the finding was only observed in two out of the 9 total replicates across the three samples, and did not perfectly mirror results from all other OIL-PCR experiments.

We feel that these results, taken with our previous OIL-PCR experiments, present compelling evidence that the observations can best be explained as a novel physical association between *Klebsiella* pneumoniae and *Romboutsia timonensis*. We have adjusted our manuscript to reflect these results.

2) Provide an additional figure with the global computational pipeline used so that readers better understand how the results were obtained.

Thank you for the suggestion. This figure was included as supplemental figure 4.

3) Provide a detailed protocol with a list of reagents and equipment.

Again, thank you for the suggestion. We have added the lab protocol to our submission and we hope that this facilitates adoption of our method.

4) Please provide a clearer comparison of your results with the HiC linkage obtained in Kent et al., rather than just the metagenomic data as it would strengthen the claims and results.

We agree that we have a unique ability to leverage our Hi-C results from experiments performed on the same samples. We have updated the manuscript to include discussion of the Hi-C results and how they compare to the OIL-PCR results.

We are happy to report that OIL-PCR confirms our previous Hi-C results. The Hi-C data and our OIL-PCR results both accurately associate the large *Klebsiella* plasmid with *Klebsiella*. Hi-C does not capture the *Romboutsia* association, but this is to be expected since we conclude that *Romboutsia* is physically associated with *Klebsiella* and does not have the genes. The Hi-C results also detect the putative transfer of the plasmid from *Klebsiella pneumoniae* to *Citrobacter braakii*. Our initial OIL-PCR results did not include this association because the two species only have a 1 SNP difference between them and were therefore clustered into the same OTU; however, when we look at SNP level differences, it does appear that *Citrobacter* is associating with all three ARGs at time points 2 and 3 only, confirming our previous Hi-C result.

5) you compare the results of sequencing with a custom database of expected sequences but what are the results if compared to the NCBI database?

In our pipeline, we use BLAST against a small database of expected target genes as a simple confirmation of our PCR product. Since the method is PCR based, with three primers for each target gene, nearly 100% of our sequences align to our small database of expected target genes. Therefore, it is not necessary to align to the NCBI database. This said, we do obtain a small novel or unexpected sequences in our PCR that do not perfectly hit our expected sequences, and are apparent as a lower identity hit. For these, as the reviewers suggest, we have used the NCBI database to determine whether the particular variant has been observed. An example of this is that we observed an OXY gene in B314, even though this was not an expected target and not included in the database.

6) The analysis of the effect of lysozyme on improving capture of gram positive bacteria is restrained to phylum. A more in-depth analysis (at the class level for instance) would be useful. For instance, it exists a clear difference between chicken and human stool samples with or without lysozyme. Why? Is it linked to some specific phylum?

Thank you for this excellent suggestion. We constrained our analysis to only show any phylum with a statistically significant difference between the two treatments; however, by restricting the results in this way, we failed to highlight global, albeit less significant, trends in the data that provide greater insight into the effects of lysozyme.

To provide a better picture of the results, we generated rarefaction curves for every taxonomic level from phylum to genus that had at least 10 OTUs (Supplemental Figure 6). We then built a phylogeny with rarefaction curves for each taxonomic level containing more than 20 OTUs as well as less abundant branches containing significant or interesting results.

We were excited to find that Lysozyme improved OTU recovery for chicken stool at every level of taxonomy observed. We also were able to show that the reduced recovery in Human stool was almost exclusively due to poor amplification of *Bacteroidaceae* which makes up the majority of human stool diversity and thus has an outsized effect on the results.